# AGALE: A Graph-Aware Continual Learning Evaluation Framework

**Tianqi Zhao**                                        *T.Zhao-1@tudelft.nl*
*Delft University of Technology*
*Delft, Netherlands*

**Alan Hanjalic**                                      *A.Hanjalic@tudelft.nl*
*Delft University of Technology*
*Delft, Netherlands*

**Megha Khosla**                                       *M.Khosla@tudelft.nl*
*Delft University of Technology*
*Delft, Netherlands*

**Reviewed on OpenReview:** *https://openreview.net/forum?id=xDTKRLyaNN*

## Abstract

In recent years, continual learning (CL) techniques have made significant progress in learning from streaming data while preserving knowledge across sequential tasks, particularly in the realm of euclidean data. To foster fair evaluation and recognize challenges in CL settings, several evaluation frameworks have been proposed, focusing mainly on the single- and multi-label classification task on euclidean data. However, these evaluation frameworks are not trivially applicable when the input data is graph-structured, as they do not consider the topological structure inherent in graphs. Existing continual graph learning (CGL) evaluation frameworks have predominantly focused on single-label scenarios in the node classification (NC) task. This focus has overlooked the complexities of multi-label scenarios, where nodes may exhibit affiliations with multiple labels, simultaneously participating in multiple tasks. We develop a graph-aware evaluation (AGALE) framework that accommodates both single-labeled and multi-labeled nodes, addressing the limitations of previous evaluation frameworks. In particular, we define new incremental settings and devise data partitioning algorithms tailored to CGL datasets. We perform extensive experiments comparing methods from the domains of continual learning, continual graph learning, and dynamic graph learning (DGL). We theoretically analyze AGALE and provide new insights about the role of homophily in the performance of compared methods. We release our framework at `https://github.com/Tianqi-py/AGALE`.

## 1 Introduction

Continual Learning (CL) describes the process by which a model accumulates knowledge from a sequence of tasks while facing the formidable challenge of preserving acquired knowledge amidst data loss from prior tasks. It finds application in several fields, such as the domain of medical image analysis, where a model has to detect timely emerging new diseases in images while maintaining the accuracy of diagnosing the diseases that have been encountered in the past. Significant achievements have been made on CL for euclidean data domains such as images and text (Aljundi et al., 2018; Parisi et al., 2018; Tang & Matteson, 2021; Hadsell et al., 2020; Van de Ven & Tolias, 2019). Recent works have also delved into the broader scenario of multi-label continual learning (MLCL) (Wang et al., 2020b; 2021; Liu et al., 2022; Wei et al., 2021), where one instance can be simultaneously associated with multiple labels.

To foster fair evaluation and identify new challenges in CL settings, several evaluation frameworks (Farquhar & Gal, 2019; Lange et al., 2023) have been proposed, focusing on the single- and multi-label classification task on the euclidean data. However, these frameworks are not trivially applicable to graph-structured data due to the complexities arising

from interconnections and multi-label nodes within graphs. Besides, existing evaluation frameworks in continual graph learning (CGL) (Ko et al., 2022; Zhang et al., 2022) evaluate the node classification task in the setting of associating nodes with a single label (which we refer to as the *single-label* scenario), thereby overlooking the possibility for nodes from previous tasks to adopt different labels in new tasks or acquire additional labels with time. For instance, in the context of a dynamically evolving social network, not only can new users with diverse interests (labels) be introduced over time, but existing users may also lose old labels or accumulate new labels continuously.

To illustrate the limitations of current CL evaluation frameworks when considering the multi-label scenario in graphs, we start with an example of a multi-label graph as in Figure 1. We use color coding to indicate the classes the nodes belong to. Please note that in what follows, we employ the term "class" to refer to the classes that correspond to a task. To refer to a class assigned to a particular node we use the term "label".

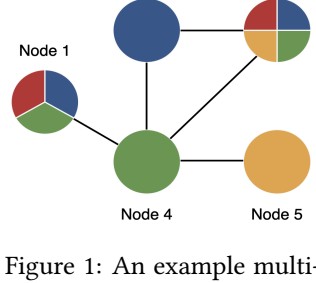

Figure 1: An example multi-label graph with colors indicating to the different node labels.

### 1.1 Limitations of current continual learning evaluation frameworks

**Lack of graph-aware data partitioning strategies.** Current experimental setups typically simulate continual learning settings by employing certain data partitioning methods over static data. However, existing CGL frameworks do not consider the multi-label scenario in the data partitioning algorithms.

The multi-label continual learning evaluation framework for Euclidean data (Kim et al., 2020a) suggest the use of *hierarchical clustering* techniques, which involves grouping classes into a single task based on their co-occurrence frequency and subsequently eliminating instances with label sets that do not align with any class groups. Applying such a technique to graph-structured data not only risks excluding nodes with a higher number of labels but also disrupts the associated topological structures.

In Figure 2, we illustrate an application of the existing MLCL framework to the multi-label graph depicted in Figure 1. The classes blue, green, and red are collectively treated as one task due to their frequent co-occurrence. Node 3, having the maximum number of labels, is removed from the graph since no task encompasses all its labels. It is noteworthy that employing hierarchical clustering techniques increases the likelihood of eliminating nodes with more labels, effectively reducing both the number of multi-labeled nodes and the associated topological structure. In the current example, proper training and testing of the model on the red class is hindered, as only one node remains in the subgraph with the label red. Besides, node 5 becomes isolated in the second subgraph.

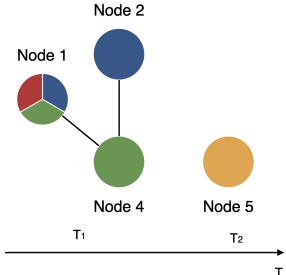

Figure 2: Subgraphs generated by grouping frequently co-occurring classes as a task.

**Generation of train/val/test sets.** Furthermore, the data partitioning algorithm is also responsible for the division of each subgraph into training, validation, and test subsets. In Figure 3 we show an example of train/val/test subsets generated using the strategy adopted by previous CGL evaluation frameworks for the task of distinguishing between blue and green classes. In particular, nodes belonging to a single class are divided independently at random into train/validation/test sets, assuming no overlap between classes. However, when each node can belong to multiple classes, an independent and random division within each class is not always feasible. For instance, the same node may serve as training data for one class and testing data for another in the same task, as is the case for node 1 in Figure 3. In this particular case, the model may observe the test data during the training process, resulting in data leakage. Conversely, considering the entire dataset as a whole for division would result in the dominance of the larger class, causing all nodes from the smaller class to be aggregated into the same split and thereby under-representing the smaller class in the other split. For instance, in multi-label datasets such as YELP, two classes can exhibit complete overlap, where all nodes from the smaller class also belong to the larger class. In this scenario, if the nodes belonging to the larger class are split first, there might be no nodes left to make the required splits for the smaller class.

**Use of predefined class orders.** Existing CGL evaluation frameworks rely on a single predefined class order in the dataset and group sets of $K$ classes as individual tasks. As the predefined order might not always reflect the true generation process of the data, it is important to evaluate CL models over several random class orders. Specifically for

multi-label scenarios, the employed class order may not only influence the nodes and their neighborhood structures presented at each time step but also affect the number of labels assigned to a particular node in a given task. We demonstrate in Figure 4, using nodes from the multi-label graph in Figure 1, how distinct class orders generate subgraphs with the same set of nodes but with different topologies and node label assignments.

**Limitations on the number of tasks.**    Last but not least, previous CGL benchmarks typically predetermined the size of model's output units, assuming an approximate count of classes in each graph during model initialization. However, this assumption is unrealistic because the eventual class size is often unknown, leading to potential inefficiencies in storage or capacity overflow.

### 1.2   Our Contributions

To tackle the above-mentioned gaps, we develop a generalized evaluation framework for continual graph learning, which is applicable both for multi-class and multi-label node classification tasks and can be easily adapted for multi-label graph and edge classification tasks. Our key contributions are as follows.

Figure 3: The split of the nodes within one subgraph generated by the previous CGL framework.

- We define two generalized incremental settings for the node classification task in the CGL evaluation framework which are applicable for both multi-class and multi-label datasets.

- We develop new data split algorithms for curating CGL datasets utilizing existing static benchmark graph datasets. We theoretically analyze the *label homophily* of the resulting subgraphs which is an important factor influencing the performance of learning models over graphs.

- We perform extensive experiments to assess and compare the performance of well-established methods within the categories of Continual Learning (CL), Dynamic Graph Learning (DGL), and Continual Graph Learning (CGL). Through our analysis, we evaluate these methods in the context of their intended objectives, identifying their constraints and highlighting potential avenues for designing more effective models to tackle standard tasks in CGL.

- We re-implement the compared models in our framework while adapting them for the unknown number of new tasks that may emerge in the future.

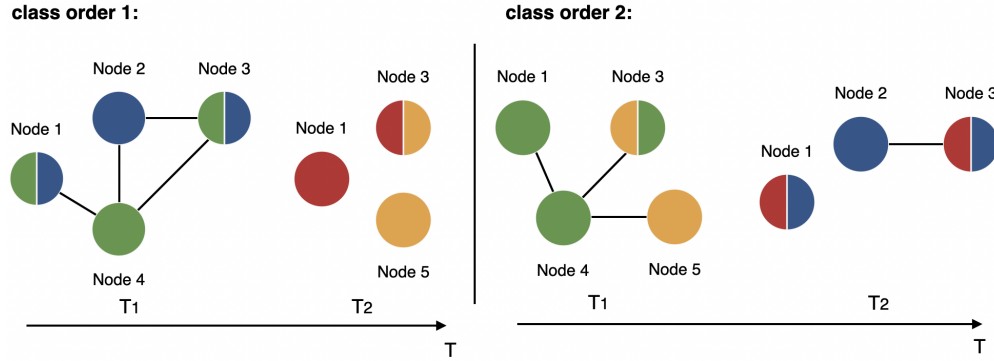

Figure 4: An example of subgraphs obtained by applying different class orders for the static multi-label graph in Figure 1.

## 2   Problem Formulation

We start by providing a general formulation of the continual learning problem for graph-structured data and elaborate on the additional complexities when the nodes in the graph may have multiple labels as compared to the single-label scenario.

**Problem Setting and Notations.** *Given a time sequence $\mathcal{T} = \{1, 2, \ldots, T\}$, at each time step $t \in \mathcal{T}$, the input is one graph snapshot $\mathcal{G}_t = (\mathcal{V}_t, \mathcal{E}_t, \mathbf{X}_t, \mathbf{Y}_t)$, with node set $\mathcal{V}_t$ and edge set $\mathcal{E}_t$. Additionally, $\mathbf{X}_t \in \mathcal{R}^{|\mathcal{V}_t| \times D}$ and $\mathbf{Y}_t \in \{0, 1\}^{|\mathcal{V}_t| \times |\mathcal{C}_t|}$ denote the feature matrix and the label matrix for the nodes in graph $\mathcal{G}_t$, where $D$ is the dimension of the feature vector, and $\mathcal{C}_t$ is the set of classes seen/available at time $t$. We assume that the node set $\mathcal{V}_t$ is partially labeled, i.e., $\mathcal{V}_t = \{\mathcal{V}_t^l, \mathcal{V}_t^u\}$, where $\mathcal{V}_t^l$ and $\mathcal{V}_t^u$ represent the labeled nodes and the unlabeled nodes in $\mathcal{G}_t$. We use $\mathbf{A}_t$ to denote the adjacency matrix of $\mathcal{G}_t$. We use $\mathcal{Y}^v$ to denote the complete label set of node $v$ and $\mathcal{Y}_t^v$ to denote the label set of node $v$ observed at time $t$.*

**Objective.** *The key objective in CGL setting, as described above, is to predict the corresponding label matrix of $\mathcal{V}_t^u$ denoted by $\mathbf{Y}_t^u$ (when the complete label set is restricted to $\mathcal{C}_t$) while maintaining the performance over the classification on nodes in all graphs in the past time steps in $\{1, 2, \ldots, t-1\}$.*

## 2.1 Differences to single-label scenario

Having explained the problem setting and the objective we now describe the key differences of the multi-label scenario as compared to the single-label case in continual graph learning, which were so far ignored by previous works resulting in various limitations as illustrated in Section 1.1.

- **Node overlaps in different tasks.** In the single-label scenario each node is affiliated with one single class, exclusively contributing to one task. The following statement, which states that no node appears in more than one task, always holds:

$$\forall i, j \in \mathcal{T}, \text{and } i \neq j, \mathcal{V}_i \cap \mathcal{V}_j = \emptyset \tag{1}$$

  However, in the multi-label scenario, one node can have multiple labels and can therefore participate in multiple tasks as time evolves. Contrary to the single-label scenario, when the nodes have multiple labels, there will exist at least a pair of tasks with at least one node in common as stated below.

$$\exists i, j \in \mathcal{T}, \text{and } i \neq j, \mathcal{V}_i \cap \mathcal{V}_j \neq \emptyset \tag{2}$$

- **Growing label sets.** In the single-label scenario, the label set of a node $v$, $\mathcal{Y}^v$, stays the same across different time steps, i.e.,

$$\forall i, j \in \mathcal{T}, \mathcal{Y}_i^v = \mathcal{Y}_j^v \tag{3}$$

  However, in the multi-label scenario, the label set of a node may grow over time, i.e., a node may not only appear in several tasks as above but also have different associated label sets, i.e., the following holds.

$$\exists i, j \in \mathcal{T}, \mathcal{Y}_i^v \neq \mathcal{Y}_j^v \tag{4}$$

- **Changing node neighborhoods.** Note that while simulating a continual learning scenario, subgraphs are curated corresponding to sets of classes/labels required to be distinguished in a particular task. In other words, the subgraph presented for a particular task only contains edges connecting nodes with the label set seen in that task. Therefore, the neighborhood of a node $v$, denoted as $\mathcal{N}^v$ can also be different across different time steps in the multi-label scenario, i.e.,

$$\exists i, j \in \mathcal{T}, \mathcal{N}_i^v \neq \mathcal{N}_j^v \tag{5}$$

In the multi-label graphs, both multi-label and single-label nodes exist, providing therefore a suitable context to develop a generalized CGL evaluation framework as elaborated in the next section.

## 3 AGALE: our evaluation framework

We present a holistic continual learning evaluation framework for graph-structured input data, which we refer to as AGALE (a graph-aware continual learning evaluation). We begin by developing two generalized incremental settings (in Section 3.1) that accommodate the requirements of the multi-label scenario (as discussed in Section 2.1) with respect to node overlaps in different tasks and growing label sets. In Section 3.2, we develop new data partitioning algorithms designed to derive subgraphs and training partitions from a static graph dataset, tailored specifically for the developed incremental settings. To underscore the significance of our approach, we provide theoretical analysis of AGALE in Section 3.3 and compare it with the previously established CGL and MLCL frameworks in Section 3.4.

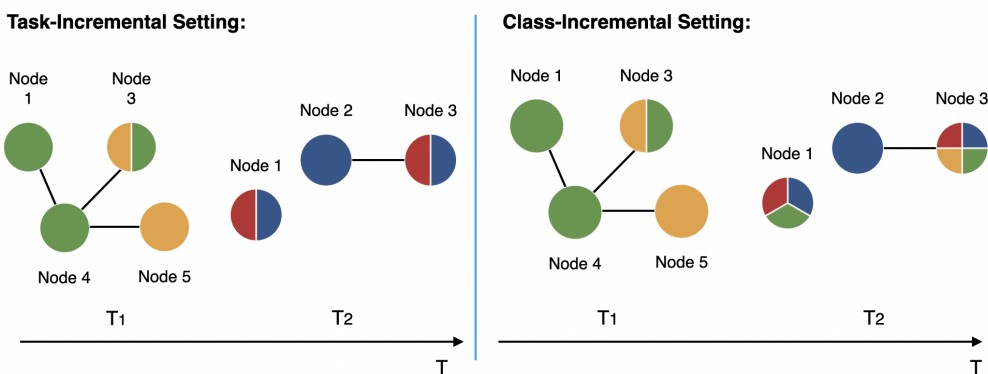

Figure 5: Visualization of our proposed generalized evaluation CGL framework AGALE.

## 3.1 Two Generalized Incremental Settings for Continual Graph Learning

We *first* define and develop two generalized incremental settings in CGL, i.e., Task-IL setting and Class-IL setting. In the **Task-IL setting**, the goal is to distinguish between classes specific to each task. Different from single-label settings, the multi-labeled nodes can be present with non-overlapping subsets of their labels in different subgraphs, as shown in Figure 5. Formally, for any node $v$ in the multi-label graph, in the Task-IL setting we have

$$\forall i, j \in \mathcal{T}, \mathcal{Y}_i^v \cap \mathcal{Y}_j^v = \emptyset.$$

In the **Class-IL setting**, the goal is to distinguish among all the classes that have been seen so far. Specifically, in addition to the same node appearing in multiple tasks as in the previous setting, a node with multiple labels can attain new labels as new tasks arrive, as shown in Figure 5. Formally, for any node $v$ in the multi-label graph,

$$\forall i, j \in \mathcal{T}, \text{if } i < j, \text{ then } \mathcal{Y}_i^v \subseteq \mathcal{Y}_j^v$$

Note that the above settings allow for node overlaps and growing/changing label sets of the same node at different time points.

## 3.2 Data Partitioning Algorithms

We now describe our data partitioning algorithms to simulate sequential data from static graphs. The design strategy of our algorithms takes into account of the node overlaps in tasks, the growing/changing label set of nodes over time, and the changing node neighborhoods while minimizing the loss of node labels and the graph's topological structure during partitioning. Our developed data partition strategy can be employed in both incremental settings and consists of the following two main parts.

- **Task sequence and subgraph sequence generation.** We employ Algorithm 1 to segment the provided graph from the dataset into coherent subgraph sequences. We first remove the classes with size smaller than a threshold $\delta$. Instead of using a predefined class order (as discussed in Section 1.1) we generate $n$ random orders of the remaining classes to simulate the random emergence of new classes in the real world. Specifically, given a dataset with $C$ classes, we group $K$ random classes as one task for one time step. At any time point $t$, let $\mathcal{C}_t$ denote the set of classes grouped for the task at time $t$. We construct a subgraph $\mathcal{G}_t = (\mathcal{V}_t, \mathcal{E}_t)$ such that $\mathcal{V}_t$ is the set of nodes with one or more labels in $\mathcal{C}_t$. The edge set $\mathcal{E}_t$ consists of the induced edges on $\mathcal{V}_t$. Note that the number of classes chosen to create one task is adaptable. In order to maximize the length of the task sequence for each given graph dataset and subsequently catastrophic forgetting, we choose $K = 2$ in this work. If the dataset has an uneven number of classes in total, the remaining last class will be grouped with the second last class group.

- **Construction of train/val/test sets.** To overcome the current limitations of generating train/val/test sets as discussed in Section 1.1, we employ Algorithm 2 to partition nodes of a given graph snapshot $\mathcal{G}_t$. For the given

subgraph $\mathcal{G}_t$, our objective is to maintain the pre-established ratios for training, validation, and test data for both the task as a whole and individual classes within the task. To achieve this, our procedure starts with the determination of the size of each class. Note that the cumulative sizes of these classes may exceed the overall number of nodes in the subgraph due to multi-labeled nodes being accounted for multiple times based on their respective labels. Subsequently, the classes are arranged in ascending order of size, starting with the smallest class. The smallest class is partitioned in accordance with the predefined proportions. Subsequent classes in the order undergo partitioning with the following steps:

- We identify nodes that have already been allocated to prior classes.
- We then compute the remaining node counts for the training, validation, and test sets in accordance with the predefined proportions for the current class.
- Finally, we split randomly the remaining nodes within the current class into train/val/test sets such that their predefined proportions are respected.

Note that for a given class order, the structural composition of each subgraph remains constant across both the incremental settings. What distinguishes these incremental settings is the label vector assigned to the nodes. Specifically, nodes with a single label manifest uniquely in one subgraph corresponding to a task. Conversely, nodes with multiple labels appear in the TASK-IL SETTING with distinct non-overlapping subsets of their original label set across various subgraphs while appearing with the expansion of their label vectors in the CLASS-IL SETTING.

---

**Algorithm 1** Task Sequence and Subgraph Sequence Generation

---

**Require:** Static graph $\mathcal{G} = (\mathcal{V}, \mathcal{E})$ with classes $\mathcal{C} = \{c_1, c_2, \ldots, c_C\}$, threshold of small classes $\delta$, group size $K$

**Ensure:** $n$ task sequences $\mathcal{S} = \{\mathcal{S}_1, \mathcal{S}_2, \ldots, \mathcal{S}_n\}$ and for each task sequence $\mathcal{S}_i$ a corresponding subgraph sequence $\mathcal{G}_i = \{\mathcal{G}_1, \mathcal{G}_2, \ldots, \mathcal{G}_T\}$

1: **for** $c_j \in \mathcal{C}$ **do**
2:      $\mathcal{V}_{c_j} = \{v_i | c_j \in y_i\}$
3: $\mathcal{C}' = \{\mathcal{C} - c_j | |\mathcal{V}_{c_j}| < \delta\}$
4: Generate $n$ random orders of $\mathcal{C}'$: $\mathcal{O} = \{\mathcal{O}_1, \mathcal{O}_2, \ldots, \mathcal{O}_n\}$
5: **for** $\mathcal{O}_j \in \mathcal{O}$ **do**
6:      **for** $t = 1$ to $\lfloor \frac{C}{k} \rfloor = T$ **do**
7:          Group the first $k$ classes as a task: $\mathcal{S}_t = \{c_1, \ldots, c_k\}$
8:          $\mathcal{O}_j = \mathcal{O}_j - \mathcal{S}_t$
9:          $\mathcal{V}_t = \{v_i | \mathbf{y_i} \cap \mathcal{S}_t \neq \emptyset\}$
10:         $\mathcal{E}_t = \{e(u, v) | e \in \mathcal{E} \land u, v \in \mathcal{V}_t\}\}$
11:         $\mathcal{G}_t = (\mathcal{V}_t, \mathcal{E}_t)$

---

In the Appendix A.1, we present an analysis of the subgraphs derived by AGALE from the given static graph in PCG as an example of showcasing the efficacy of our approach.

### 3.3 Theoretical Analysis Of AGALE

As studied in previous works (Ma et al., 2021; Zhao et al., 2023), the similarity of labels between neighboring nodes (usually termed label homophily) influences the performance of various graph machine learning algorithms for the task of node classification in the static case. We here provide a theoretical analysis of AGALE with respect to the label homophily of generated subgraphs under different conditions. We would later use our theoretical insights and the dataset properties to analyze the performance of various methods. We use the following definition of label homophily for multi-label graphs proposed in Zhao et al. (2023).

**Definition 1.** *Given a multi-label graph $\mathcal{G}$, the label homophily $h$ of $\mathcal{G}$ is defined as the average of the Jaccard similarity of the label set of all connected nodes in the graph:*

$$h = \frac{1}{|\mathcal{E}|} \sum_{(i,j) \in \mathcal{E}} \frac{|\mathcal{Y}^i \cap \mathcal{Y}^j|}{|\mathcal{Y}^i \cup \mathcal{Y}^j|}$$

---

**Algorithm 2** Train and Test Partition Algorithm Within One Subgraph

---

**Require:** subgraph $\mathcal{G}_t$ in subgraph sequence $\{\mathcal{G}_1, \mathcal{G}_2, \ldots, \mathcal{G}_T\}$, proportion set $P$ for train, validation, and test $P = \{P_{train}, P_{val}, P_{test}\}$

**Ensure:** the split within subgraph $\mathcal{G}_t = \{\mathcal{V}_t^{train}, \mathcal{V}_t^{val}, \mathcal{V}_t^{test}\}$ for task $\mathcal{S}_t$

1: Get the classes for the current task $\mathcal{S}_t = \{c_1, \ldots, c_k\}$
2: $\mathcal{O}' = Sort_{ascend}(|\mathcal{V}_{c_j}|)$ for $c_j \in \mathcal{S}_t$
3: initialize empty node set $\mathcal{V}_t^{train}, \mathcal{V}_t^{val}$, and $\mathcal{V}_t^{test}$
4: initialize empty encountered nodes set $\mathcal{V}_t$
5: **for** $c \in \mathcal{O}'$ **do**
6:      $\mathcal{V}_c = \{v_i | c \in y_i\}$
7:      **if** $c$ is the smallest class in $\mathcal{S}_i$ **then**
8:          Randomly split $\mathcal{V}_c$ into $\mathcal{V}_c^{train}, \mathcal{V}_c^{val}, \mathcal{V}_c^{test}$ according to $P$
9:      **else**
10:          Calculate the size of train/val/test set $|\mathcal{V}_c^{train}|, |\mathcal{V}_c^{val}|, |\mathcal{V}_c^{test}|$ according to $P$
11:          $\mathcal{V}_t^{dup} = \mathcal{V}_c \cap \mathcal{V}_t$
12:          $\mathcal{V}_c = \mathcal{V}_c - \mathcal{V}_t^{dup}$
13:          **for** $v_i \in \mathcal{V}_{dup}$ **do**
14:              **for** $split \in [\mathcal{V}_c^{train}, \mathcal{V}_c^{val}, \mathcal{V}_c^{test}]$ **do**
15:                  **if** $v_i$ in $split$ **then**
16:                      $|split| = |split| - 1$
17:          **for** $split \in [\mathcal{V}_c^{train}, \mathcal{V}_c^{val}, \mathcal{V}_c^{test}]$ **do**
18:              Randomly choose $|split|$ nodes from $\mathcal{V}_c$ to add to $split$
19:      add $\mathcal{V}_c^{train}, \mathcal{V}_c^{val}, \mathcal{V}_c^{test}$ to $\mathcal{V}_t^{train}, \mathcal{V}_t^{val}, \mathcal{V}_t^{test}$
20:      add $\mathcal{V}_c$ to $\mathcal{V}_t$

---

Let for any two connected nodes $i, j \in \mathcal{V}$, $h_{\mathcal{G}}^{e(i,j)}$ denotes the label homophily over the edge $e(i,j) \in \mathcal{E}$ in graph $\mathcal{G}$. We then have the following result about the label homophily of $e(i,j)$ in the subgraph $\mathcal{G}_t$ generated by AGALE at time $t$.

**Theorem 1.** *For any edge $e(i,j) \in \mathcal{E}$ and any subgraph at time $t$, $\mathcal{G}_t$ such that $e(i,j) \in \mathcal{E}_t$, $h_{\mathcal{G}_t}^{e(i,j)} \geq h_{\mathcal{G}}^{e(i,j)}$ when at least one of the nodes in $\{i,j\}$ is single-labeled. For the case when both nodes $i, j$ are multi-labeled, we obtain $h_{\mathcal{G}_t}^{e(i,j)} \geq h_{\mathcal{G}}^{e(i,j)}$ with probability at least $(1 - (1 - h_{\mathcal{G}}^{e(i,j)})^K)$ for TASK-IL SETTING and $(1 - (1 - h_{\mathcal{G}}^{e(i,j)})^{Kt})$ for CLASS-IL SETTING.*

*Proof.* In the multi-label graphs, one pair of connected nodes belongs to the following three scenarios: 1) two single-labeled nodes are connected, 2) a single-label node is connected to a multi-labeled node, and 3) two multi-labeled nodes are connected.

**Scenario 1:** Note that at any time step $t$ two nodes $i$ and $j$ are connected if and only if at least one label for each node appears in $\mathcal{C}_t$. As in the first scenario, both the nodes are single-labeled, and the label homophily score for edge $e(i,j)$ stays the same in the subgraph as in the original graph:

$$h_{\mathcal{G}_t}^{e(i,j)} = h_{\mathcal{G}}^{e(i,j)} = \begin{cases} 0, & \text{if } \mathcal{Y}^i \neq \mathcal{Y}^j \\ 1, & \text{if } \mathcal{Y}^i = \mathcal{Y}^j \end{cases} \tag{6}$$

**Scenario 2:** In the second scenario, where one single-labeled node $i$ is connected to a multi-labeled node $j$, at any time step $t$, when $e(i,j)$ appears in the subgraph $\mathcal{G}_t$,

$$h_{\mathcal{G}_t}^{e(i,j)} \geq h_{\mathcal{G}}^{e(i,j)} \begin{cases} h_{\mathcal{G}_t}^{e(i,j)} = h_{\mathcal{G}}^{e(i,j)} = 0, & \text{if } \mathcal{Y}^i \notin \mathcal{Y}^j \\ h_{\mathcal{G}_t}^{e(i,j)} = \begin{cases} \frac{1}{2}, & \text{if } \mathcal{Y}^i \subset \mathcal{C}_t \cap \mathcal{Y}^j \\ 1, & \text{if } \mathcal{C}_t \cap \mathcal{Y}^j = \mathcal{Y}^i \end{cases} \geq h_{\mathcal{G}}^{e(i,j)}, & \text{if } \mathcal{Y}^i \in \mathcal{Y}^j \end{cases} \tag{7}$$

Combining equation 6 and equation 7 we note that when at least one node in an edge is single-labeled, the label homophily of the corresponding edge will be equal to more than that in the static graph, thereby completing the first part of the proof.

**Scenario 3:** In the third scenario, where two multi-labeled nodes $i$ and $j$ are connected, at any time step $t$, when $e(i,j)$ appears in the subgraph $\mathcal{G}_t$, it holds $\mathcal{C}_t \cap \mathcal{Y}_i \neq \emptyset$ and $\mathcal{C}_t \cap \mathcal{Y}_j \neq \emptyset$. In this scenario, the label homophily of an edge depends on the relationship between $\mathcal{Y}_i \cap \mathcal{Y}_j$ and $\mathcal{C}_t$:

$$
\begin{cases}
0 = h_{\mathcal{G}_t}^{e(i,j)} < h_{\mathcal{G}}^{e(i,j)} & \text{if } \mathcal{Y}^i \cap \mathcal{Y}^j \cap \mathcal{C}_t = \emptyset \\
\begin{cases} h_{\mathcal{G}_t}^{e(i,j)} = \frac{1}{2} & \text{\textsc{Task-IL setting}} \\ h_{\mathcal{G}_t}^{e(i,j)} \geq \frac{1}{2t} & \text{\textsc{Class-IL setting}} \end{cases} & \text{if } \mathcal{Y}^i \cap \mathcal{Y}^j \neq \emptyset, \mathcal{Y}^i \cap \mathcal{Y}^j \cap \mathcal{C}_t \subset \mathcal{Y}^i \cap \mathcal{Y}^j, \mathcal{Y}^i \cap \mathcal{Y}^j \cap \mathcal{C}_t \subset \mathcal{C}_t \\
h_{\mathcal{G}_t}^{e(i,j)} \geq h_{\mathcal{G}}^{e(i,j)} & \text{if } \mathcal{Y}^i \cap \mathcal{Y}^j \subset \mathcal{C}_t \\
h_{\mathcal{G}_t}^{e(i,j)} = 1 \geq h_{\mathcal{G}}^{e(i,j)} & \text{if } \mathcal{C}_t \subseteq \mathcal{Y}^i \cap \mathcal{Y}^j
\end{cases}
\tag{8}
$$

Note that all the statements hold in both incremental settings except for the second condition, where $\mathcal{Y}_t^i \cap \mathcal{Y}_t^j \cap \mathcal{C}_t$ is the strict subset of $\mathcal{Y}_t^i \cap \mathcal{Y}_t^j$ and $\mathcal{C}_t$. With a relatively smaller size of $|\mathcal{C}_t| = K = 2$ in our setting, we have in the \textsc{Task-IL setting}, $|\mathcal{Y}_t^i \cap \mathcal{Y}_t^j| = 1$ and $|\mathcal{Y}_t^i \cup \mathcal{Y}_t^j| = 2$:

$$
h_{\mathcal{G}_t}^{e(i,j)} = \frac{|\mathcal{Y}_t^i \cap \mathcal{Y}_t^j|}{|\mathcal{Y}_t^i \cup \mathcal{Y}_t^j|} = \frac{1}{2}
\tag{9}
$$

while in the \textsc{Class-IL setting}, because $|\mathcal{Y}_t^i \cap \mathcal{Y}_t^j| \geq 1, |\mathcal{Y}_t^i \cup \mathcal{Y}_t^j| \leq Kt$, we obtain

$$
h_{\mathcal{G}_t}^{e(i,j)} = \frac{|\mathcal{Y}_t^i \cap \mathcal{Y}_t^j|}{|\mathcal{Y}_t^i \cup \mathcal{Y}_t^j|} \geq \frac{1}{2t}
\tag{10}
$$

We can now upper bound the probability of the worst case event, i.e., when an edge $e(i,j)$ exists at time $t$ but $\mathcal{C}_t \cap \mathcal{Y}^i \cap \mathcal{Y}^j = \emptyset$. This can only happen if the classes in set $\mathcal{C}_t$ are chosen from the set $\mathcal{Y}^i \cup \mathcal{Y}^j \setminus \mathcal{Y}^i \cap \mathcal{Y}^j$. For \textsc{Task-IL setting}, the probability of choosing at least one element of $\mathcal{C}_t$ from the common labels of node $i$ and $j$ is equal to $h_{\mathcal{G}}^{e(i,j)}$. Then the probability that none of the classes in $\mathcal{C}_t$ appear in the common set is at most $(1 - h_{\mathcal{G}}^{e(i,j)})^{|\mathcal{C}_t|}$. The proof is completed by noting the fact that $|\mathcal{C}_t| = K$ for \textsc{Task-IL setting} and $|\mathcal{C}_t| = Kt$ for \textsc{Class-IL setting} at time step $t$. $\square$

### 3.4 Comparison With Previous Evaluation Frameworks

In response to overlooked challenges in established CGL and MLCL evaluation frameworks, as detailed in Section 1, our framework tackles these issues by the following.

- **Incorporation of the multi-label scenario.** Contrary to previous evaluation frameworks AGALE accommodates single-label and multi-label node nodes in the following ways.

  - For single-label nodes, our framework expands upon previous methods during the task sequence's creation phase. It introduces dynamics in label correlations by allowing random class ordering to generate the task sequence. This results in diverse subgraph sequences, mimicking the random emergence of new trends in the real world.
  - Regarding the multi-label scenario, as shown in Figure 5, our framework allows for update/change of label assignments for a given node in the \textsc{Task-IL setting} and expansion of the node's label set in the \textsc{Class-IL setting}.

- **Information preservation and prevention of data leakage**

  - As described in Section 3.2, the data partitioning strategies of AGALE ensure that no nodes from the original multi-label static graph are removed while creating the tasks. Single-labeled nodes appear once in the task

sequence in both settings, while multi-labeled nodes surface with different labels in TASK-IL SETTING and CLASS-IL SETTING. Specifically, they appear with non-overlapping subsets of their label set in TASK-IL SETTING, and as the class set expands, their entire label set is guaranteed to be seen by the model before the final time step in CLASS-IL SETTING.

– Previous CGL evaluation frameworks split the nodes into train and evaluation sets within each class, not considering the situation where one node can belong to multiple classes in the task. Such a strategy may lead to data leakage as one node can be assigned to training and testing sets for the same task. During task training on a subgraph comprising various classes, our framework ensures no overlap among the training, validation, and test sets. Single-labeled nodes exclusively belong to one class, preventing their re-splitting after the initial allocation. For multi-label nodes that have been allocated to a particular class (see lines 11 and 12 in Algorithm 2), we exclude them from the remaining nodes of other classes they belong to, eliminating any potential data leakage during training and evaluation within one subgraph.

– In addition, we approach the continual learning setting by not allowing the inter-task edges. This deliberate choice means that, upon the arrival of a new task, the model no longer retains access to the data from the previous time steps.

- **Ensuring fair split across different classes and the whole graph.** Due to the differences in the class size, a split from the whole graph will result in the bigger class dominating the splits, leaving the small class underrepresented in the splits. Moreover, the split within each class may result in data leakage in one subgraph, as explained in the previous paragraph. To maintain a fair split despite differences in class sizes, our framework prioritizes splitting smaller classes initially. It subsequently removes already split nodes from larger class node sets. This approach guarantees an equitable split within each class and from within the whole subgraph, preventing larger classes from dominating the splits and ensuring adequate representation for smaller classes.

- **Application for graph/edge-level CGL.** AGALE can be directly applied for the graph classification task, each input data is an independent graph without interconnections. For the edge classification task, our framework can be applied by first transforming the original graph $\mathcal{G}$ into a line graph $L(\mathcal{G})$, where for each edge in $\mathcal{G}$, we create a node in $L(\mathcal{G})$; for every two edges in $\mathcal{G}$ that have a node in common, we make an edge between their corresponding nodes in $L(\mathcal{G})$.

## 4 Related Work

### 4.1 Continual Learning

Continual Learning (van de Ven & Tolias, 2019; Hadsell et al., 2020; Nguyen et al., 2018; Aljundi et al., 2019; Li & Hoiem, 2016; Aljundi et al., 2017; Wang et al., 2023a), a fundamental concept in machine learning, addresses the challenge of enabling models to learn from and adapt to evolving data streams over time. Continual learning has applications in a wide range of domains, including computer vision, natural language processing, and reinforcement learning, making it an active area of research with practical implications for the lifelong adaptation of machine learning models. Unlike traditional batch learning, where models are trained on static datasets, continual learning systems aim to learn from new data while preserving previously acquired knowledge sequentially. This paradigm is particularly relevant in real-world scenarios where data is non-stationary and models need to adapt to changing environments.

The key objectives of continual learning are to avoid catastrophic forgetting, where models lose competence in previously learned tasks as they learn new ones, and to ensure that the model's performance on earlier tasks remains competitive. Various techniques have been proposed in the literature to tackle these challenges, which can be categorized into four categories.

- **Knowledge distillation methods.** The methods from this category (Li & Hoiem, 2016; Wang et al., 2021; 2020b) retain the knowledge from the past by letting the new model mimic the old model on the previous task while adapting to the new task. Overall, the learning objective can be summarized as to minimize the following loss function:

$$\mathcal{L} = \lambda_o \mathcal{L}_{\text{old}}\left(\mathbf{Y}_o, \hat{\mathbf{Y}}_o\right) + \mathcal{L}_{\text{new}}\left(\mathbf{Y}_n, \hat{\mathbf{Y}}_n\right) + \mathcal{R}, \tag{11}$$

where $\mathcal{L}_{\text{old}}$ and $\mathcal{L}_{\text{new}}$ represent the loss functions corresponding to the old and new tasks, respectively. The parameter $\lambda_o$ is the weight for balancing the losses, and $\mathcal{R}$ encapsulates the regularization term. The process of

transferring knowledge from a pre-existing model (teacher) to a continually evolving model (student) in knowledge distillation unfolds within $\mathcal{L}_{\text{old}}$, where the new model undergoes training to align its predictions on new data for the old task, denoted as $\hat{\mathbf{Y}}_o$, with the predictions of the previous model on the same new data for the old task, represented as $\mathbf{Y}_o$. Simultaneously, the new model approximates its prediction of the new data on the new task $\hat{\mathbf{Y}}_n$ to their true labels $\mathbf{Y}_n$. For example, LwF (Li & Hoiem, 2016) minimize the difference between the outputs of the previous model and the new model on the new coming data for the previous tasks while minimizing the classification loss of the new model on the new task.

- **Regularization strategies.** The methods in this category maintain the knowledge extracted from the previous task by penalizing the changes in the parameters $\theta$ of the model trained for the old tasks. Typically, the following loss is minimized:

$$\mathcal{L}(\theta) = \mathcal{L}_{\text{new}}(\theta) + \lambda \sum_i \Omega_i \left(\theta_{\mathbf{i}} - \theta_{\mathbf{i}}^*\right)^2 \tag{12}$$

where $\mathcal{L}_{\text{new}}$ denotes the loss function for the new task, $\theta$ is the set of model parameters. The parameter $\lambda$ functions as the weight governing the balance between the old and new tasks, while $\Omega_i$ represents the importance score assigned to the $ith$ parameter $\theta_{\mathbf{i}}$. For example, MAS (Aljundi et al., 2017) assigns importance scores for the parameters by measuring how sensitive the output is to the change of the parameters. The term $\theta_i^*$ refers to the prior task's parameter determined through optimization for the previous task.

- **Replay mechanisms.** Methods from this category extract representative data from the previous tasks and employ them along with the new coming data for training to overcome catastrophic forgetting (Shin et al., 2017; Kim et al., 2020b). Methods under this category mainly differ with respect to their approaches to sampling representative data from the old task for storage in the buffer. For example, Kim et al. (2020b) maintains a target proportion of different classes in the memory to tackle the class imbalance in the multi-label data.

- **Dynamic architectures.** Methods from this category (Lee et al., 2017; Wei et al., 2021) dynamically expand their architecture when needed for new tasks. This expansion may include adding new layers or neurons to accommodate new knowledge. For example, Lee et al. (2017) dynamically expands the network architecture based on the relevance between new and old tasks.

Another line of work in CL focuses on benchmarking evaluation methods. For instance, Farquhar & Gal (2019) and Lange et al. (2023) provide more robust and realistic evaluation metrics for the CL methods, incorporating real-world challenges like varying task complexities and the stability gap.

## 4.2 Continual Graph Learning

As a sub-field of continual learning, Continual Graph Learning (CGL) addresses the catastrophic forgetting problem as the model encounters new graph-structured data over time. Within CGL, two primary lines of work exist. The first involves establishing evaluation frameworks that define incremental settings in CGL scenarios and their corresponding data partitioning algorithms. The second line of work focuses on proposing new methods based on specific predefined CGL incremental settings derived from these evaluation frameworks. Our work mainly falls into the first category in which we develop a more holistic evaluation framework covering the multi-label scenario for graph-structured data.

The previously established CGL frameworks focus on benchmarking tasks in CGL. For instance, Zhang et al. (2022) defined Task- and Class- Incremental settings for single-labeled node and graph classification tasks in CGL and studied the impact of including the inter-task edges among the subgraph. Ko et al. (2022) expanded this by adding the domain- and time-incremental settings and including the link prediction task in the CGL benchmark. Additionally, surveys like Febrinanto et al. (2022) and Yuan et al. (2023) focus on categorizing the approaches in CL and CGL.

However, none of the above works sufficiently addressed the complexities of defining the multi-label node classification task within the CGL scenario. The only exception is Ko et al. (2022), which used a graph with multi-labeled nodes, but that too in a domain incremental setting. In particular, each task was constituted of nodes appearing from a single domain. Consequently, a node appears in one and only one task together with all of its labels. This does not cover the general multi-label scenario in which the same node can appear in multiple tasks each time with different or expanding label sets.

Existing methods for CGL focus mainly on the multi-class scenario and fall into one of the four categories (see the previous subsection) of continual learning methods. For example, GRAPHSAIL (Xu et al., 2020) is a knowledge distillation approach that distills each node's local and global structure and its self-embedding knowledge, respectively. Regularization approach TWP (Liu et al., 2020) adds a penalization to the parameters that are important to the learned topological information in addition to the task-related loss to stabilize the parameters playing pivotal roles in the topological aggregation. ERGNN (Zhou & Cao, 2021) is based on the replay mechanism and carefully selects nodes from the old tasks to the buffer and replays them with the new graph. Wang et al. (2020a) combines replay and regularization to preserve existing patterns.

## 4.3 Learning on dynamic graphs

Since streaming graphs find applications in various domains, including social network analysis, recommendation systems, fraud detection, and knowledge graph refinement, several methods (Wang et al., 2023b; Yu et al., 2018; 2017; Xu et al., 2019) have been proposed in the field of dynamic graph learning (DGL) to utilize the knowledge from the past to enhance the model's performance on the graph in the current timestamp. For example, Rossi et al. (2020) uses the memory unit to represent the node's history in the compressed format, and Pareja et al. (2019) uses recurrent architecture between the models trained for the adjacent time steps to let the new model inherent knowledge extracted from the old tasks. However, the designing goal of the methods in DGL is to utilize the knowledge extracted from the old tasks to enhance the performance of the model on the current task, while in CGL, we focus on the catastrophic forgetting problem, i.e., the model needs not only to perform well on the current task but also on the previous tasks in the task sequence. We compare and analyze the models from these two categories in detail in Section 6.

## 4.4 Application of graph machine learning in continual learning

Some work (Tang & Matteson, 2021; Liu et al., 2023) also attempts to use graph structures to alleviate catastrophic forgetting in Euclidean data. For instance, Tang & Matteson (2021) augments independent image data in memory with a learnable random graph, capturing similarities among them to alleviate catastrophic forgetting. However, as our current focus is solely on graph-structured data, these endeavors fall beyond the scope of this study.

# 5 Experiment Setup

In this section, we test the state-of-art models from CL, DGL, and CGL domains. Note that in this study, we employ $P = 3$, indicating that we generate three random orders for the classes in each dataset in the experimental section. We introduce the models according to their categories.

## 5.1 Methods

This subsection introduces all the methods used in the experiment section. The CL methods use Graph Convolutional Network (GCN) (Kipf & Welling, 2016) as the backbone.

- **SIMPLEGCN**: We train GCN on each of the subgraph sequences without any continual learning technique, which is denoted as SIMPLEGCN in the following sections.

- **JOINTTRAINGCN**: We also include GCN trained on all the tasks simultaneously and therefore should not have the catastrophic forgetting problem. This setting is referred to as JOINTTRAINGCN in the following section.

- **Continual Learning Methods**: We choose Learning Without Forgetting (LwF), Elastic Weight Consolidation (EWC), and Memory Aware Synapses (MAS) from this category. LwF distill the knowledge from the old model to the new model to prevent the model from catastrophic forgetting. EWC and MAS are both regularization-based methods. The difference is that EWC penalizes the changes in the parameters that are important to the previous task, while MAS measures the importance of the parameters based on the sensitivity of the output on the parameters.

- **Dynamic Graph Neural Network**: We choose EVOLVEGCN (Pareja et al., 2019) from this category, which uses recurrent architecture between the models trained for the adjacent time steps to let the new model inherent knowledge extracted from the old tasks to enhance the model's performance on the current task.

- **Continual Graph Learning Methods**: We choose ERGNN (Zhou & Cao, 2021) from this category, which samples representative nodes from the old tasks in the buffer and replays them with the new data to address the catastrophic forgetting problem.

## 5.2   Datasets

We demonstrate our evaluation framework on 3 multi-label datasets in this work. We also include 1 multi-class dataset CoraFull as an example to demonstrate the generalization of our evaluation framework on single-label nodes. We include the description of the CoraFull and the results on it in the Appendix A.2.

The inter-task edges are defined in (Zhang et al., 2022) as the edges that connect the new subgraph to the overall graph. We do not allow inter-task edges in our evaluation framework, i.e., at each time step, only the subgraph for the new task is used as input. The reason is that in CL, the assumption is that the model loses access to the data from the previous time steps. With the inter-task edges, the node features from the previous time step would also be used as input, which violates this assumption and alleviates the forgetting problem.

Below, we introduce the datasets used in this work:

1. PCG(Zhao et al., 2023), in which nodes are proteins and edges correspond to the protein functional interaction, and the labels the phenotype of the proteins.

2. DBLP(Akujuobi et al., 2019), in which nodes represent authors and edges the co-authorship between the authors, and the labels indicate the research areas of the authors.

3. Yelp(Zeng et al., 2019), in which nodes correspond to the customer reviews and edges to their friendships with node labels representing the types of businesses.

The statistics about the datasets are summarized in Table 1. We use the label homophily defined for multi-label graphs in Zhao et al. (2023). Following the application of a data partitioning algorithm, the given static graphs by the datasets are split into subgraph sequences. We also summarize the characteristics of the subgraphs to provide insights into the partitioned structure.

Table 1: The data statistics. Specifically, $|\mathcal{V}|$, $|\mathcal{E}|$, $|\mathcal{C}|$, $\overline{|\mathcal{L}|}$, and $r_{homo}$ denote the number of nodes, edges, classes, mean label count per node, and label homophily of the static graph given by the dataset, respectively. $|T|$ signifies the count of tasks in the resulting task sequence. Additionally, $\overline{|\mathcal{V}|}$ and $\overline{|\mathcal{E}|}$ represent the average number of nodes and edges in a subgraph. Further details on label homophily are captured through $\overline{|r|}_{tsk}$ and $\overline{|r|}_{cls}$, representing the averaged label homophily of subgraphs in the Task-IL setting and Class-IL setting), respectively.

| | $|\mathcal{V}|$ | $|\mathcal{E}|$ | $|\mathcal{C}|$ | $\overline{|\mathcal{L}|}$ | $|T|$ | $r_{homo}$ | $\overline{|\mathcal{V}|}$ | $\overline{|\mathcal{E}|}$ | $\overline{|r|}_{tsk}$ | $\overline{|r|}_{cls}$ |
|---|---|---|---|---|---|---|---|---|---|---|
| PCG | $3K$ | $37K$ | 15 | 1.93 | 7 | 0.17 | 808 | 4763 | 0.64 | 0.38 |
| DBLP | $28K$ | $68K$ | 4 | 1.18 | 2 | 0.76 | $15K$ | $37K$ | 0.86 | 0.81 |
| Yelp | $716K$ | $7.34M$ | 100 | 9.44 | 50 | 0.22 | $121K$ | $921K$ | 0.75 | 0.47 |

In Theorem 1 we theoretically analyzed the label homophily of the edges in the subgraphs where we showed that in cases of single-labeled nodes and for higher homophily edges, the homophily in subgraphs typically increases. Table 1 further shows that the average label homophily of the subgraphs is in fact higher than the label homophily of the corresponding static graph.

## 5.3   Evaluation

### 5.3.1   Metrics

We evaluate the models using performance matrix $\mathbf{M} \in \mathbb{R}^{T \times T}$, where $\mathbf{M}_{i,k}$ denotes the performance score reported by an evaluation metric (e.g. AUC-ROC, average precision etc.) on task $\mathcal{S}_k$ after the model has been trained over a sequence of tasks from $\mathcal{S}_1$ to $\mathcal{S}_i$. At each time step $t$, the average performance of the model is measured by the

average of the model's performances on task $\mathcal{S}_1$ to task $\mathcal{S}_i$, i.e., the average of the row $i$ in performance matrix $\mathbf{M}$. After the whole task sequence is presented to the model, we report the average performance $AP$ as:

$$AP = \frac{\sum_{i=1}^{T} \mathbf{M}_{T,i}}{T} \tag{13}$$

which is the higher, the better.

We use the average forgetting $AF$ score proposed in Lopez-Paz & Ranzato (2017). The forgetting on task $\mathcal{S}_i$ is measured by the performance change on task $\mathcal{S}_i$ after the model is trained on the whole task sequence. Formally, we report the average forgetting $AF$ on all the tasks as:

$$AF = \frac{\sum_{i=1}^{T} (\mathbf{M}_{T,i} - \mathbf{M}_{i,i})}{T - 1} \tag{14}$$

Note that we here compute a single metric to quantify the incurred forgetting over past tasks when the model is trained for the last task $\mathcal{S}_T$. The summand indicates the performance decrease on some task $\mathcal{S}_i$ after learning on later task $\mathcal{S}_T$.

When the average forgetting is negative, its absolute value indicates the averaged performance decrease on all previous tasks when the model is trained on the last task $\mathcal{S}_T$ in the task sequence.

A positive AF score indicates that the performance on some of the past tasks actually increased after training on task $\mathcal{S}_T$. A positive AF score might be the result of correlation among tasks that the model exploited, thus showing an improvement over past tasks.

Such an observation may be when the tasks from a graph are highly correlated with each other, training on the new task would help further improve the performance on the old tasks.

Overall, we report the $AP$ and $AF$ for each model, and the scores we obtain from the two metrics are interpreted in the following Table 2.

|          | high $AF$ | low $AF$ |
|----------|-----------|----------|
| high $AP$ | preserves well-rounded knowledge across all the tasks | performs well on the new task, while forgetting about the old tasks |
| low $AP$ | preserves the knowledge from the old tasks and harms the overall performance indicates the tasks are not correlated, improvements on one task harm the performance on the other tasks | forgets about the old task, and fail to perform well on the new task |

Table 2: The interpretation of the average performance score ($AP$) and the average forgetting score ($AF$).

### 5.3.2 Visualization

We use the heatmaps and lineplots to visualize the performance matrix $\mathbf{M}$. Due to the limited space, we add the heatmaps in the Appendix A.4. The lineplots are shown in Figure 7, which have the time steps as $x$ axis, the $y$ axis indicates the average performance of the model over all the tasks that have been encountered so far.

## 6 Results and Analysis

In this section, we summarize the experimental results on the multi-label datasets in the TASK-IL SETTING and CLASS-IL SETTING defined in section 2 in the Table 3 and Table 4, respectively. To use a single numerical value to quantify the overall performance of the models, we calculate an average performance matrix $\hat{\mathbf{M}}$ from the performance matrices from the three random splits and report the $AP$ and $AF$ from the averaged performance matrix.

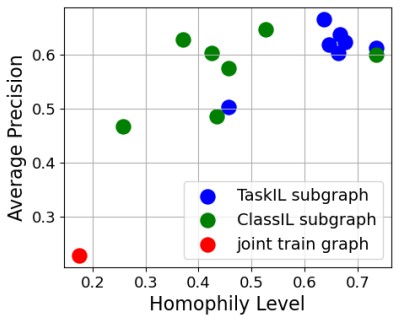

(a) The visualization of the performances of GCN on the subgraphs and the joint train graph from PCG and the label homophily of the graphs.

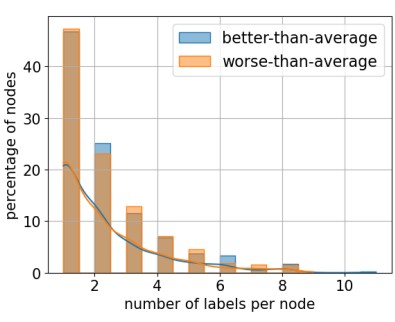

(b) The distribution of the number of labels per node in the better-than-average subset and in the worse-than-average subset.

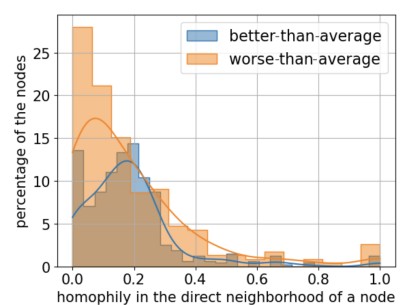

(c) The distribution of the label homophily of the nodes in the better-than-average subset and in the worse-than-average subset.

Figure 6: Visualization of the analysis on the performance of SIMPLEGCN and JOINTTRAINGCN using PCG as an example.

## 6.1 Lower and Upper Bounds in CGL

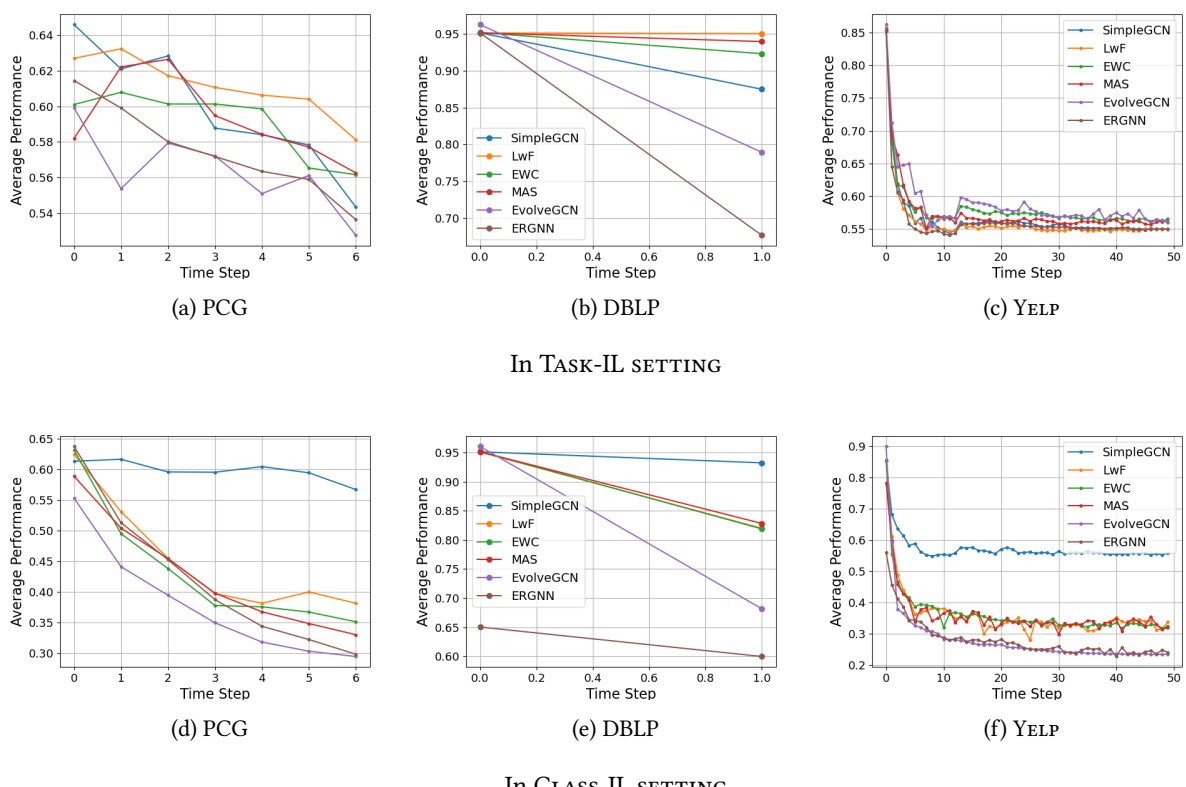

Figure 7: Learning curves showing the dynamics of the average performance during learning on the task sequences of different datasets. The color coding and legend names remain consistent across all subfigures. To avoid obstructing the line plot, we omit the legend in the subplots corresponding to PCG.

In the previous CGL frameworks (Zhang et al., 2022; Ko et al., 2022), SIMPLEGCN and JOINTTRAINGCN are shown to have the worst and the best performance. Such a result is also expected as (i) SIMPLEGCN is employed on sequential

data without any enhanced abilities to deal with catastrophic forgetting (thereby showing performance degradation) and (ii) in JointTrainGCN all data is used to train the base GNN. However, the results from multi-label datasets in both incremental settings, as shown in Table 3 and Table 4, reveal that SimpleGCN and JointTrain are no longer suitable as lower and upper bounds for evaluating CGL performance in a more generalized scenario of multi-label datasets. In the following, we theoretically and empirically analyze the rationale behind such a finding.

### 6.1.1 Label homophily and GCN

GNNs, specifically GCN, which is used as a base network are known to have better performance on high label-homophilic graphs. As shown in Theorem 1, splitting labels into distinct prediction tasks and creating subgraphs for each task results in an increase in label homophily of the edges in the subgraphs as compared to that in the full graph. In particular, if in a dataset there are a large number of single-labeled nodes in the full graph with a non-zero edge label homophily, the increase in label homophily of edges in subgraphs helps SimpleGCN to assign correct labels to the corresponding nodes. However, in JointTrain, the presence of diverse neighborhoods around single-labeled nodes leads to low label homophily, impacting its performance negatively.

**Empirical evidence.** Figure 6 illustrates the above statements with an example from one random shuffle of PCG using the subgraphs generated for the Task-IL setting (colored in blue), Class-IL setting (colored in green) and the original static graph given by the dataset (colored in red). On the $x$ axis, we show the label homophily level of the input graphs, while on the $y$ axis, we show the performance of SimpleGCN after it is trained on the subgraph in the corresponding incremental settings and JointTrainGCN on the whole static graph. We make the following observations.

- The subgraphs in Task-IL setting and Class-IL setting have higher label homophily than the full graph, explaining the better performance of SimpleGCN as compared to JointTrainGCN.

- We also observe that as compared to Task-IL setting, the subgraphs generated for Class-IL setting have lower label homophily. This happens because of expanding label sets in Class-IL setting.

In Figure 6b and 6c, we further analyze the causes of the bad performance of the JointTrainGCN. We used the JointTrainGCN model on test nodes from the joint train graph in PCG and calculated an average precision score for each node. The mean value of the scores is then used as a threshold to divide the test nodes into the set of nodes that perform *better-than-average* and the *worse-than-average* performing node subset, indicated by the blue and orange bars in the plots. To remove the influence of the difference in the sizes of the subsets, we use the percentage of the nodes in the corresponding subset as the $y$ axis.

Based on the edge homophily defined in 1, we define the label homophily in the direct neighborhood of a node as the averaged edge homophily connected to this node:

**Definition 2.** *For a node $v$ in the graph $\mathcal{G}$, we define the label homophily of a node $v$ with respect to its immediate neighborhood $\mathcal{N}^v$, represented as $h^v$, as the average of label homophily of the edges connected to $v$:*

$$h^v = \frac{\sum_{e(i,j)|j \in \mathcal{N}^v} h^{e(i,j)}}{|\mathcal{N}^v|}$$

We make the following observations.

- In Figure 6b, the percentage of single-labeled nodes in the worse-than-average performing subset is higher than that the better-than-average subset.

- Figure 6c shows that in fact, the high percentage of worse-performing nodes have very low label homophily (computed using Definition 2) close to 0.

- The above two observations indicate that the performance of JointTrainGCN suffers due to the presence of a higher percentage of low label homophily edges with at least one single-labeled node.

For completeness, we include in Figure 6c a Kernel Density Estimation on the node homophily distribution, which shows a clear shift in the distributions of the label homophily in the better-performing subset as compared to the worse-than-average subset.

In the following sections, we summarize the performance of the chosen baselines in the TASK-IL SETTING and CLASS-IL SETTING and provide a detailed analysis of the performances of the baselines on different datasets.

## 6.2 Results in TASK-IL SETTING

Table 3: Performance of the baseline models in the TASK-IL SETTING setting. The performances are reported in Average Precision. "AP" stands for Average Precision, and the higher, the better. "AF" indicates the average forgetting, and the higher, the better.

| TASK-IL | PCG | | DBLP | | YELP | |
|---|---|---|---|---|---|---|
| | AP | AF | AP | AF | AP | AF |
| SIMPLEGCN | $54.34 \pm 0.04$ | $-6.11 \pm 0.03$ | $87.47 \pm 0.12$ | $-15.76 \pm 0.00$ | $54.87 \pm 0.03$ | $-1.43 \pm 0.05$ |
| LwF | $58.12 \pm 0.05$ | $-2.84 \pm 0.02$ | $95.01 \pm 0.01$ | $-0.98 \pm 0.00$ | $54.89 \pm 0.03$ | $-2.07 \pm 0.05$ |
| EWC | $56.17 \pm 0.03$ | $-3.77 \pm 0.03$ | $92.28 \pm 0.05$ | $-6.51 \pm 0.01$ | $56.53 \pm 0.05$ | $-0.17 \pm 0.02$ |
| MAS | $56.26 \pm 0.04$ | $-2.64 \pm 0.03$ | $93.93 \pm 0.03$ | $-3.17 \pm 0.00$ | $56.05 \pm 0.03$ | $-0.76 \pm 0.05$ |
| EVOLVEGCN | $52.76 \pm 0.06$ | $-3.68 \pm 0.03$ | $78.94 \pm 0.25$ | $-35.20 \pm 0.00$ | $55.93 \pm 0.07$ | $-5.11 \pm 0.07$ |
| ERGNN | $53.64 \pm 0.06$ | $-1.39 \pm 0.02$ | $67.70 \pm 0.03$ | $-24.96 \pm 0.00$ | $54.99 \pm 0.03$ | $-0.92 \pm 0.04$ |
| JOINTTRAIN | $22.47 \pm 0.47$ | $-$ | $85.60 \pm 0.25$ | $-$ | $13.80 \pm 0.08$ | $-$ |

Table 3 presents results for three real-world multi-label datasets in TASK-IL SETTING. In general, the knowledge distillation method LwF excels on graphs with shorter task sequences (e.g., PCG and DBLP with 7 and 2 tasks, respectively). In contrast, all methods perform comparably on the graph with a long task sequence in YELP with 50 tasks, among which regularization-based methods like EWC and MAS slightly outperform other approaches. This disparity arises because LwF distills knowledge only from the last time step, leading to a performance drop with longer sequences. Meanwhile, regularization-based methods, like EWC and MAS, which penalize the changes in the important parameters for previous tasks, prove effective for longer task sequences. The weak performance of the replay-based methods ERGNN indicates the importance of including the local topological structure around the nodes in the buffer instead of sampling isolated nodes in the buffer. Dynamic graph neural networks like EVOLVEGCN struggle with substantial forgetting despite achieving notable average precision scores because they only focus on the current task. We visualize the learning curve of the models in the TASK-IL SETTING on PCG, DBLP, and YELP in Figure 7a, 7b, and 7c, respectively. The $x$ axis indicates the current time step, and the corresponding value on the $y$ axis infers the average performance of the model at the current time step over all the tasks encountered so far.

## 6.3 Detailed analyses on different datasets

**PCG.** PCG has a relatively shorter task sequence with 7 tasks. SIMPLEGCN showcases competitive scores but is susceptible to forgetting, indicating the low correlation among the tasks. LwF outperforms SIMPLEGCN and notably improved robustness against forgetting, which indicates the shorter task sequence in PCG contributes to the effectiveness of LwF in retaining task knowledge because LwF only distills knowledge from the previous model. EWC and MAS also exhibit competitive performance, demonstrating moderate resistance to forgetting. Meanwhile, because of the low correlations among the tasks, EVOLVEGCN faces challenges using with a lower performance and notable forgetting. JOINTTRAINGCN has the poorest performance because of the low label homophily level on the joint train graph.

**DBLP.** DBLP has the shortest task sequence length with only 2 tasks. LwF once again stands out with the highest performance and minimal forgetting. The SIMPLEGCN has the worst forgetting on DBLP compared to the other two datasets, indicating the tasks in DBLP have the lowest task correlation. While EWC and MAS present comparable performance to LwF, they suffer from worse forgetting. Notably, the low task correlation also results in the low performance and extreme forgetting of EVOLVEGCN and ERGNN, indicating the information from the previous task

that lies in the model, and the data can not assist the model's performance on the new task. And because the joint train graph in DBLP has the highest level of label homophily, the JointTrainGCN also achieves better performances compared to its performance on the other two multi-label datasets.

**Yelp.** The Yelp dataset is characterized by the longest task sequence encompassing 50 tasks and featuring the highest task correlations, which is indicated by the competitive performance shown by SimpleGCN. Despite the extended task sequence, training on a new task does not significantly impair performance on the previous tasks. The long task sequence poses a potential challenge for LwF, as prolonged sequences lead to increased forgetting. EWC and MAS emerge as robust performers in this demanding setting, demonstrating solid performance with competitive performances and modest forgetting. EvolveGCN encounters a lower score coupled with considerable forgetting, as the high task correlation makes the utilization of the previous model helpful to improve the performance of the current task, EvolveGCN pays no attention to maintaining the performance on the old tasks. Additionally, ERGNN achieves a comparable performance with minimal forgetting, positioning it as a strong contender on the Yelp dataset. JointTrainGCN achieves the lowest performance because of the low label homophily level in the joint train graph.

### 6.4 Class-IL setting

Table 4: Performance of the baseline models in Class-IL setting setting. The performances are reported in Average Precision. "AP" stands for Average Precision and the higher the better. "AF" indicates the average forgetting, and the higher, the better.

| Class-IL | PCG | | DBLP | | Yelp | |
|---|---|---|---|---|---|---|
| | AP | AF | AP | AF | AP | AF |
| SimpleGCN | $56.70 \pm 0.04$ | $-2.48 \pm 0.03$ | $93.22 \pm 0.03$ | $-3.90 \pm 0.00$ | $55.78 \pm 0.03$ | $-1.56 \pm 0.05$ |
| LwF | $38.13 \pm 0.13$ | $3.48 \pm 0.03$ | $81.98 \pm 0.18$ | $-0.69 \pm 0.00$ | $33.72 \pm 0.05$ | $0.59 \pm 0.05$ |
| EWC | $35.12 \pm 0.13$ | $2.17 \pm 0.03$ | $81.88 \pm 0.06$ | $-8.92 \pm 0.00$ | $32.32 \pm 0.08$ | $1.71 \pm 0.03$ |
| MAS | $33.00 \pm 0.15$ | $0.98 \pm 0.02$ | $82.82 \pm 0.10$ | $-4.87 \pm 0.00$ | $32.07 \pm 0.07$ | $-1.10 \pm 0.04$ |
| EvolveGCN | $29.44 \pm 0.12$ | $0.23 \pm 0.01$ | $68.18 \pm 0.03$ | $-29.69 \pm 0.00$ | $23.45 \pm 0.06$ | $-0.70 \pm 0.05$ |
| ERGNN | $29.80 \pm 0.14$ | $-2.82 \pm 0.03$ | $59.99 \pm 0.12$ | $3.14 \pm 0.00$ | $24.00 \pm 0.06$ | $-0.12 \pm 0.01$ |
| JointTrain | $22.47 \pm 0.47$ | — | $85.60 \pm 0.25$ | — | $13.80 \pm 0.08$ | — |

Table 4 presents results for three real-world multi-label datasets in Class-IL setting. Overall, SimpleGCN achieves a superior performance across all datasets. This performance contrast is noteworthy when compared to its performance on multi-class datasets in the previous works (Zhang et al., 2022; Ko et al., 2022). The key distinction lies in our evaluation framework, where we enable the label vectors of multi-labeled nodes to expand during Class-IL setting. In essence, this approach incorporates the previous labels of multi-labeled nodes as part of the target labels in subsequent tasks. This strategy serves a dual purpose: it mitigates the problem of forgetting while simultaneously improving the performance on earlier tasks. This improvement is indicated by the positive average forgetting scores in the Class-IL setting context. The performance of JointTrainGCN is not influenced by the change in the setting, as it is trained on all the tasks simultaneously.

The drop in the performances of other baseline models is a result of the increasing number of classes in the tasks at each time step, i.e., the difficulty of the task increases at each time step. We visualize the learning curve of the models in the Class-IL setting on PCG, DBLP, and Yelp in Figure 7d, 7e, and 7f, respectively. The $x$ axis indicates the current time step, and the corresponding value on the $y$ axis infers the average performance of the model at the current time step over all the tasks encountered so far. Below, we analyze the performance of the chosen baseline models on each of the datasets.

### 6.5 Detailed analyses on different datasets

**PCG.** SimpleGCN leads with the highest average performance overall tasks with low forgetting, as it has no CL technique to prevent forgetting. On the other hand, the CL methods sacrificed the average performance on the task sequence but successfully maintained a positive AF. This means the knowledge distillation- and regularization-based models are able to retain the knowledge from the old tasks in the Class-IL setting. EvolveGCN and ERGNN achieve

comparable average performance on the task sequence, but the ERGNN fails to retain the knowledge from the old task as it only samples the isolated nodes in the replay buffer while ignoring the topological structure. JOINTTRAINGCN remains the worst-performing model because of the low label homophily in the input graph.

**DBLP.** DBLP has the shortest task sequence, but SIMPLEGCN and CL methods LwF, EWC, and MAS suffered from the most severe forgetting problem on it. These negative average forgetting scores observed in DBLP indicate low task correlation, i.e., the knowledge from the old task hinders the model from achieving better performance on the new task. As the least multi-labeled graph, DBLP witnesses the least pronounced performance dip in the CLASS-IL SETTING compared to TASK-IL SETTING. This observation suggests that multi-label datasets pose a more challenging test for models when the label vectors of nodes continue to grow.

**YELP.** In YELP, nodes are more multi-labeled compared to PCG and DBLP, as shown in Table 1. Overall, we see a clear performance difference in CLASS-IL SETTING compared to TASK-IL SETTING on YELP. Furthermore, knowledge distillation- and regularization-based methods surpass the dynamic graph neural network EVOLVEGCN and the replay-based method ERGNN. This is primarily due to the fact that EVOLVEGCN neglects the preservation of knowledge from previous tasks, which ultimately hampers overall performance. ERGNN, on the other hand, disregards the topological structure surrounding the sampled experience nodes, further impacting its efficacy in handling the evolving tasks.

## 7    Conclusion

We develop a new evaluation framework which we refer to as AGALE for continual graph learning. Filling in the gaps in the current literature, we (i) define two generalized incremental settings for the more general multi-label node classification task, (ii) develop new data split algorithms for curating CGL datasets, and (iii) perform extensive experiments to evaluate and compare the performance of methods from continual learning, dynamic graph learning and continual graph learning. Through our theoretical and empirical analyses we show important differences of the multi-label case with respect to the more studied single-label scenario. We believe that our work will encourage the development of new methods tackling the general scenario of multi-label classification in continual graph learning.

Following the current literature, we focus on quantifying catastrophic forgetting in AGALE. In realistic scenarios, there is also the case where the model could be required to selectively forget about the past. For example, users in the social network might not further show interest in certain topics and un-follow some of the friends. Developing new evaluation metrics as well as new models to reward selective forgetting of some tasks while avoiding catastrophic forgetting overall is an interesting avenue for future research.

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

# A    Appendix

**Organization.** We analyze the characteristics of the subgraphs generated by AGALE and compare them with the full graph given in the dataset in Section A.1. Furthermore, we also apply our AGALE on single-label graph CoraFull and summarize and analyze the results in section A.2 to further demonstrate the generalization of AGALE in sing-label scenarios. Additionally, we provide detailed time and space complexity analysis in Section A.3 and measure and summarize the run time of the conducted experiments as well. Last but not least, we provide the visualization of the performance matrix using heatmaps in Section A.4.

## A.1    Data Analysis Of The Subgraphs

In this section, we present an analysis of the subgraphs derived by our evaluation framework from the static graph in PCG, showcasing the efficacy of our approach. Figure 8 illustrates the degree distribution of nodes within the seven subgraphs generated from the PCG dataset. We see from the degree distribution that the nodes in the subgraphs also have a similar degree distribution to the nodes in the original static graph.

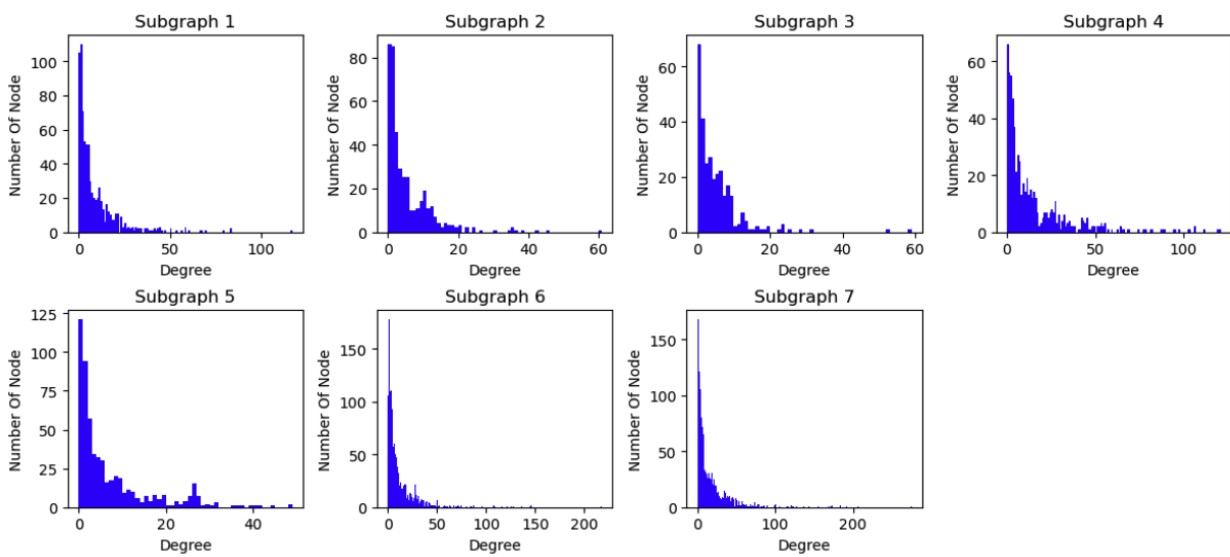

Figure 8: The Node Degree Distribution In the seven Subgraphs Generated From PCG.

## A.2    Application Of Our Evaluation Framework On Single-label Graphs

In this section, we provide an example of applying our evaluation framework to single-label graphs. Here, we use CoraFull as an example. We summarize the characteristics of CoraFull in Table 5. As shown in the Table, CoraFull has 70 classes, which are divided into 35 tasks in 3 random orders.

In Table 6 and 7, we summarize the performance of LwF and ERGNN on the dataset CoraFull in Task-IL setting and Class-IL setting and use the line plots in Figure 9 to visualize the learning curves of the chosen models in the two settings on CoraFull.

Table 5: The data statistics. Specifically, $|\mathcal{V}|$, $|\mathcal{E}|$, $|\mathcal{C}|$, $\overline{|\mathcal{L}|}$, and $r_{homo}$ denote the number of nodes, edges, classes, mean label count per node, and label homophily of the static graph given by the dataset, respectively. $|T|$ signifies the count of tasks in the resulting task sequence. Additionally, $\overline{|\mathcal{V}|}$ and $\overline{|\mathcal{E}|}$ represent the average number of nodes and edges in a subgraph. Further details on label homophily are captured through $\overline{|r|}_{tsk}$ and $\overline{|r|}_{cls}$, representing the averaged label homophily of subgraphs in the TASK-IL SETTING and CLASS-IL SETTING), respectively.

|  | $|\mathcal{V}|$ | $|\mathcal{E}|$ | $|\mathcal{C}|$ | $|T|$ | $r_{homo}$ | $\overline{|\mathcal{V}|}$ | $\overline{|\mathcal{E}|}$ | $\overline{|r|}_{tsk}$ | $\overline{|r|}_{cls}$ |
|---|---|---|---|---|---|---|---|---|---|
| CORAFULL | $19K$ | $130K$ | 70 | 35 | 0.57 | 566 | 1035 | 0.99 | 0.99 |

Table 6: Performance of the baseline models in TASK-IL SETTING setting. The performances are reported in Average Precision. "AP" stands for Average Precision and the higher the better. "AF" indicates the average forgetting, and the higher, the better.

| TASK-IL SETTING | CORAFULL | |
|---|---|---|
|  | AP | AF |
| LwF | $53.46 \pm 0.12$ | $-9.53 \pm 0.16$ |
| ERGNN | $59.49 \pm 0.20$ | $4.37 \pm 0.34$ |

Table 7: Performance of the baseline models in CLASS-IL SETTING setting. The performances are reported in Average Precision. "AP" stands for Average Precision and the higher the better. "AF" indicates the average forgetting, and the higher, the better.

| CLASS-IL SETTING | CORAFULL | |
|---|---|---|
|  | AP | AF |
| LwF | $5.42 \pm 0.15$ | $-7.45 \pm 0.14$ |
| ERGNN | $40.39 \pm 0.27$ | $-56.08 \pm 0.25$ |

### A.3 Time And Space Complexity Analysis

Here, we provide theoretical time and space complexity analysis of the models used in this work.

**Complexity analysis for the base model.** As the base model used by all compared methods is GCN (Kipf & Welling, 2016), we first analyze its complexity. To keep the notations simpler let us assume that the feature (including the input features) dimension in all layers is equal to $d$. Let $n, m$ denote the number of nodes and edges, respectively in the input graph at any time point. For the sake of brevity in the presentation, we assume that the number of nodes and edges stay the same for all time points.

For GCN, at each layer, the operation includes feature transformation, neighbourhood aggregation, and activation. The feature transformation over two layers leads to the multiplication of matrices of sizes (i) $n \times d$ and $d \times d$, and (ii) $d \times d$ and $d \times d$ which leads to a total time complexity of $\mathcal{O}(nd^2)$.

And the neighborhood aggregation requires a multiplication between matrices of size $n \times n$ and $n \times d$, yielding $\mathcal{O}(n^2 d)$. In practice, we compute this using a sparse operator, such as the PyTorch scatter function for each entry $(i, j)$ in the adjacency matrix of the edge $e \in \mathcal{E}$, which yields a total cost of $\mathcal{O}(md)$. Finally, the activation is an element-wise function with the time complexity of $\mathcal{O}(n)$. Overall, the time complexity of a $L$ layer GCN is $\mathcal{O}(nd^2 L + mdL + nL)$.

For computing space requirements of GCN, we include (i) the space required for the input adjacency matrix of size $n \times n$, (ii) the feature matrix of size $n \times d$, and (iii) the model itself with $d^2 + d$ parameters for weight and bias in each layer. In total, the space complexity of GCN is $\mathcal{O}(n^2 + nd + L(d^2 + d))$.

As all methods mentioned in this work either use GCN as the backbone model or are built upon GCN, we denote in the following discussion and the Table 8 the time and space requirement of GCN as $T_{GCN}$ and $S_{GCN}$ respectively.

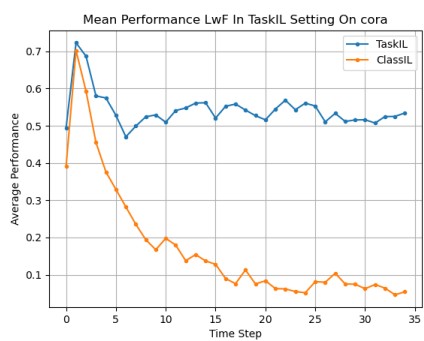

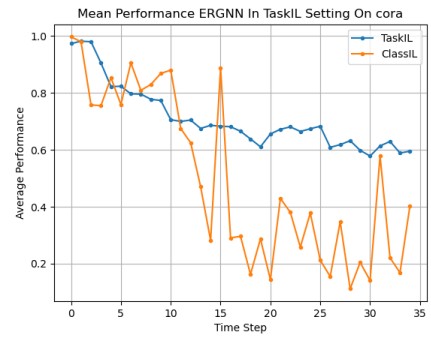

(a) LwF in TASK-IL SETTING and CLASS-IL SETTING on CORAFULL

(b) ERGNN in TASK-IL SETTING and CLASS-IL SETTING on CORA-FULL

Figure 9: Our Framework on Single-label Datasets

**Complexity analysis of SIMPLEGCN.**    SIMPLEGCN trains a GCN for each time step $t \in \mathcal{T}$. And because it does not apply any continual learning techniques to remember from previous time steps, the time is the same with GCN, i.e., $\mathcal{O}(|\mathcal{T}|T_{GCN})$ and the space complexity is $S_{GCN}$.

**Complexity analysis of LwF.**    LwF uses GCN as the backbone model, the GCN is trained at each time step $t \in \mathcal{T}$ for the new task, which gives the time complexity of $\mathcal{O}(|\mathcal{T}|T_{GCN})$. To do the knowledge distillation, the previous model also calls GCN forward passes with time complexity of $\mathcal{O}(|\mathcal{T}|T_{GCN})$. Overall, the time complexity is $\mathcal{O}(2|\mathcal{T}|T_{GCN})$. The space consumption consists of loading the current GCN model and the previous GCN model for the knowledge distillation, with space complexity of $\mathcal{O}(2S_{GCN})$.

**Complexity analysis of EWC.**    EWC calls forward passes of GCN at each time step, and for each parameter at each time step $t \in \mathcal{T}$, the values on the diagonal of the fisher matrix are approximated using the value of each model parameter itself and its gradient. This is an element-wise calculation, which gives the time complexity of $\mathcal{O}(|\mathcal{T}| \times M)$, $M$ indicates the number of parameters in the GCN, which is $L(d^2 + d)$. Overall, the time complexity of EWC is the time complexity of GCN plus the calculation of the fisher matrix, which is $\mathcal{O}(|\mathcal{T}| \times (T_{GCN} + M))$. The space requirement of EWC consists of the space requirement of GCN, and the matrix stores the gradients of the parameters at each time step of size $\mathcal{O}(|\mathcal{T}| \times M)$. In total, the space complexity of EWC is $\mathcal{O}(S_{GCN} + |\mathcal{T}|M)$.

**Complexity analysis of MAS.**    Similarly, MAS using GCN as backbone model, at each time step $t \in \mathcal{T}$, there are forward passes of GCN and the calculation of fisher matrix for parameters, which gives the overall time complexity of $\mathcal{O}(|\mathcal{T}|(T_{GCN} + M))$. The space requirement of MAS consists of the space complexity of GCN and one matrix for the gradient of the parameters of size $\mathcal{O}(M)$, which yields in total $\mathcal{O}(S_{GCN} + M)$.

**Complexity analysis of EVOLVEGCN.**    EVOLVEGCN is a method from the category of Dynamic Graph Neural networks, which trains a new model at each time step with time complexity same as GCN, i.e., $T_{GCN}$ and updates the model parameter using a recurrent neural network using the corresponding parameter from previous time step as input, which has the time complexity of $\mathcal{O}(|\mathcal{T}|nd)$. Overall, EVOLVEGCN is more expensive than the other Continual Learning methods with time complexity of $\mathcal{O}(T_{GCN} + |\mathcal{T}|nd)$. The space requirement of EVOLVEGCN consists of the space requirement of the GCN plus the recurrent unit for the reset, update, and new gate with $3(d^2 + d)$. In our implementation, we use one recurrent layer for each layer in GCN. Thus the overall space complexity yields $\mathcal{O}(2S_{GCN} + 3M)$.

**Complexity analysis of ERGNN.**    ERGNN is a replay-based method. At each time step $t \in \mathcal{T}$, it retrains the GCN on the current graph and the buffer-nodes-formed graph, and the sampling process goes through the nodes in the new graph, which gives the time complexity of $\mathcal{O}(2|\mathcal{T}|T_{GCN} + n)$. The space requirement of ERGNN consists of the buffer size of $|\mathcal{B}|$ and the space complexity of GCN. In total, it yields the space complexity of $\mathcal{O}(S_{GCN} + |\mathcal{B}|)$.

**Complexity analysis of JOINTTRAINGCN.** JOINTTRAINGCN has the same time and space complexity as the base model GCN. It uses the whole static graph as input and is only trained once without the task sequence.

The above time and space complexity analyses are summarized in Table 8.

Table 8: The simplified time complexity analysis. $T_{GCN}$ and $S_{GCN}$ corresponds to the time and space requirement of the base GCN model.

| Model | Time Complexity | Space Complexity |
|---|---|---|
| SIMPLEGCN | $\mathcal{O}(|\mathcal{T}|T_{GCN})$ | $S_{GCN}$ |
| LwF | $\mathcal{O}(2|\mathcal{T}|T_{GCN})$ | $\mathcal{O}(2S_{GCN})$ |
| EWC | $\mathcal{O}(|\mathcal{T}| \times (T_{GCN} + M))$ | $\mathcal{O}(S_{GCN} + |\mathcal{T}|M)$ |
| MAS | $\mathcal{O}(|\mathcal{T}|(T_{GCN} + M))$ | $\mathcal{O}(S_{GCN} + M)$ |
| EVOLVEGCN | $\mathcal{O}(T_{GCN} + |\mathcal{T}|nd)$ | $\mathcal{O}(2S_{GCN} + 3M)$ |
| ERGNN | $\mathcal{O}(2|\mathcal{T}|T_{GCN} + n)$ | $\mathcal{O}(S_{GCN} + |\mathcal{B}|)$ |
| JOINTTRAINGCN | $T_{GCN}$ | $S_{GCN}$ |

Besides, we also measured the run time of the experiments in this work. The results are summarized in Table 9. Note that the running time of the experiments can be biased due to different splits and how the resources are distributed on the computer. The theoretical analysis may provide more insights into the complexity of time.

Table 9: The computation time of the experiments from Section 6 in second. The computation time is measured with one random split for each dataset.

| | TASK-IL SETTING | | | CLASS-IL SETTING | | |
|---|---|---|---|---|---|---|
| | PCG | DBLP | YELP | PCG | DBLP | YELP |
| SIMPLEGCN | 43.47 | 801.85 | 77163.32 | 88.31 | 886.79 | 104869.58 |
| LwF | 49.17 | 1193.40 | 142468.01 | 111.23 | 732.76 | 264657.47 |
| EWC | 51.32 | 939.79 | 79804.48 | 135.44 | 790.73 | 200649.31 |
| MAS | 148.19 | 1169.50 | 75255.31 | 135.93 | 1230.88 | 145917.67 |
| EVOLVEGCN | 40.34 | 427.99 | 120580.88 | 94.94 | 497.80 | 310540.41 |
| ERGNN | 47.27 | 536.20 | 416090.74 | 172.05 | 131.57 | 167624.39 |
| JOINTTRAINGCN | 166.82 | 796.19 | 80827.72 | 166.82 | 796.19 | 80827.72 |

## A.4 Visualization of the Performance Matrix

In this section, we provide the visualization of the performance matrix using the heatmap on the three multi-label datasets. In the heatmap, each cell corresponds to a unique entry in $\mathbf{M}$, and its position in the heatmap mirrors its position in the matrix. We use the gradient of the color to indicate the performance. The color intensity indicates the magnitude of the value.

In the Figure 10, 11, and 12, we show the heatmaps correspond to the performance matrices of the baseline models in the TASK-IL SETTING on datasets PCG, DBLP, and YELP, respectively, while in the Figure 13, 14, and 15, we show the heatmaps correspond to the performance matrices of the baseline models in the CLASS-IL SETTING on datasets PCG, DBLP, and YELP, respectively.

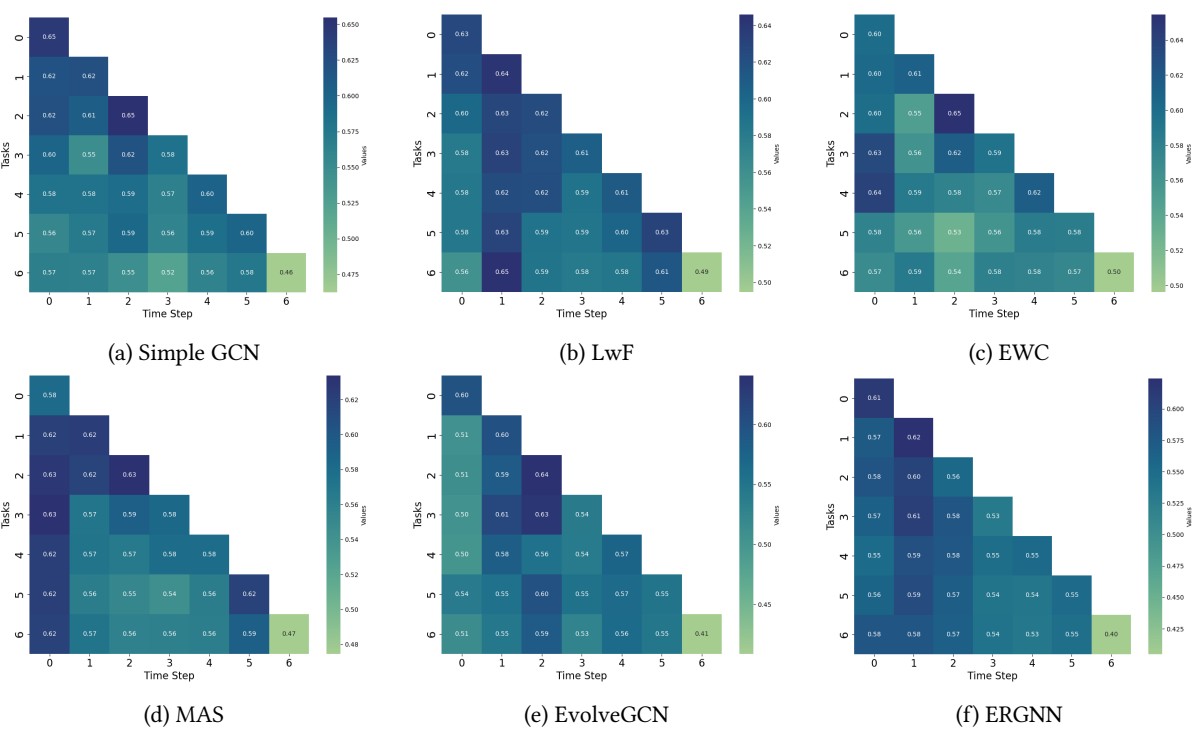

Figure 10: Visualization of the performance matrix of the methods in TaskIL setting on dataset PCG

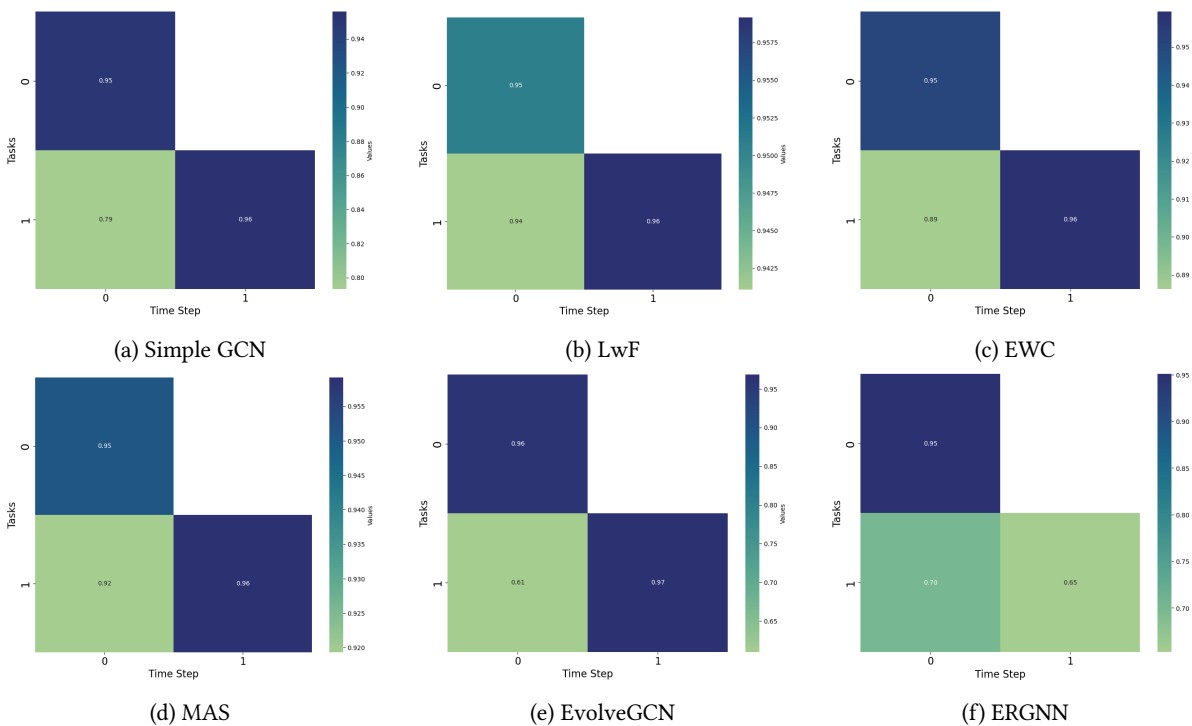

Figure 11: Visualization of the performance matrix of the methods in TaskIL setting on dataset DBLP

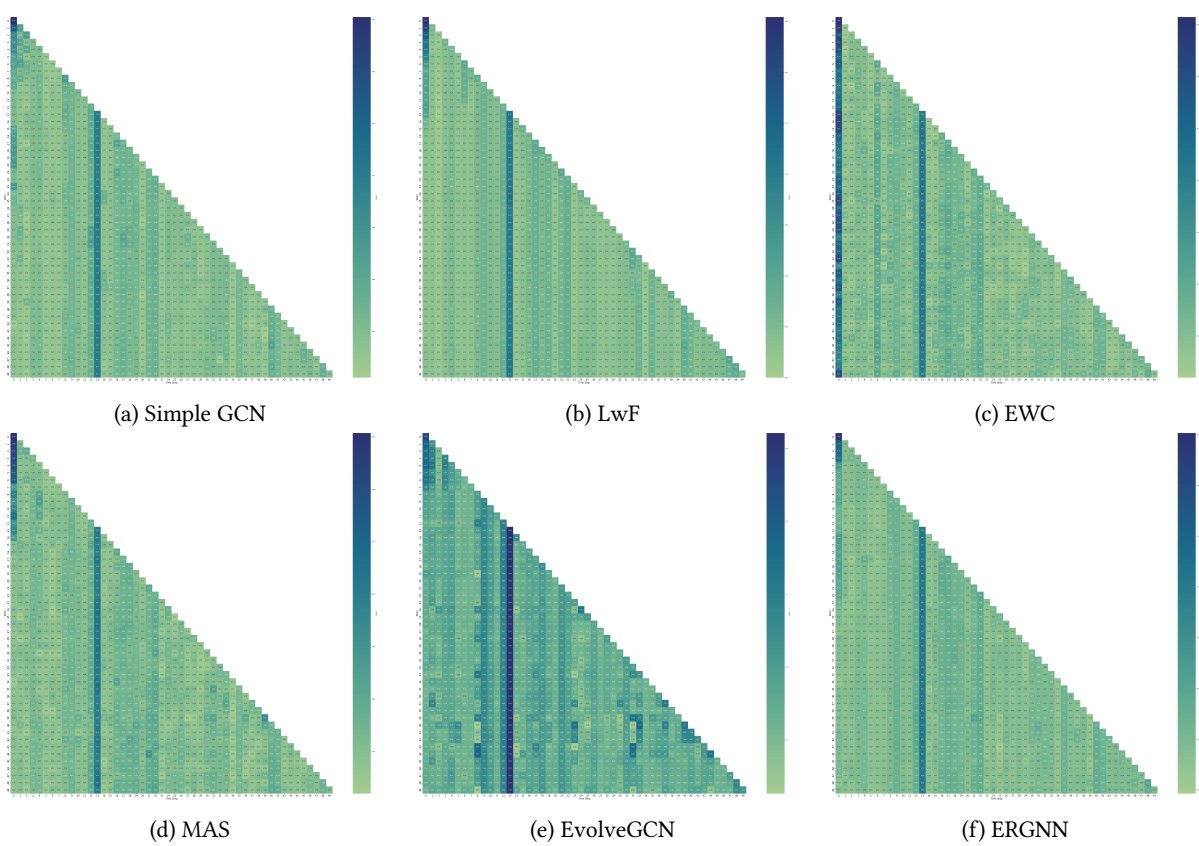

Figure 12: Visualization of the performance matrix of the methods in TaskIL setting on dataset Yelp

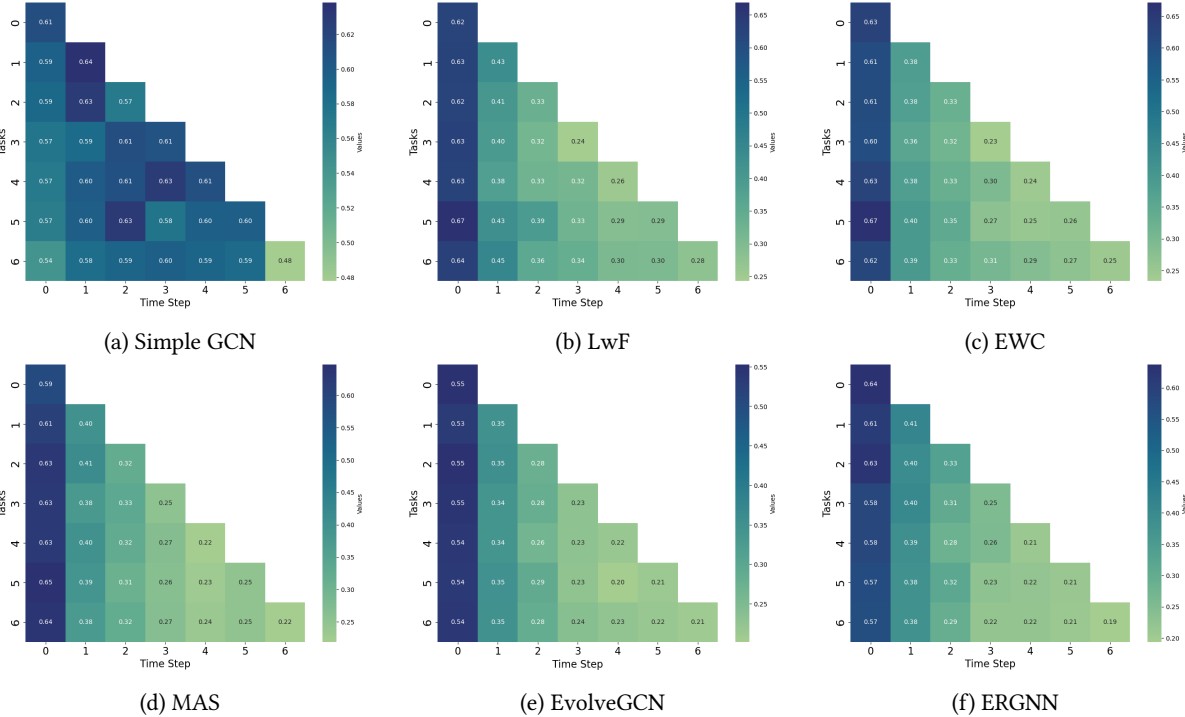

Figure 13: Visualization of the performance matrix of the methods in Class-IL setting on dataset PCG

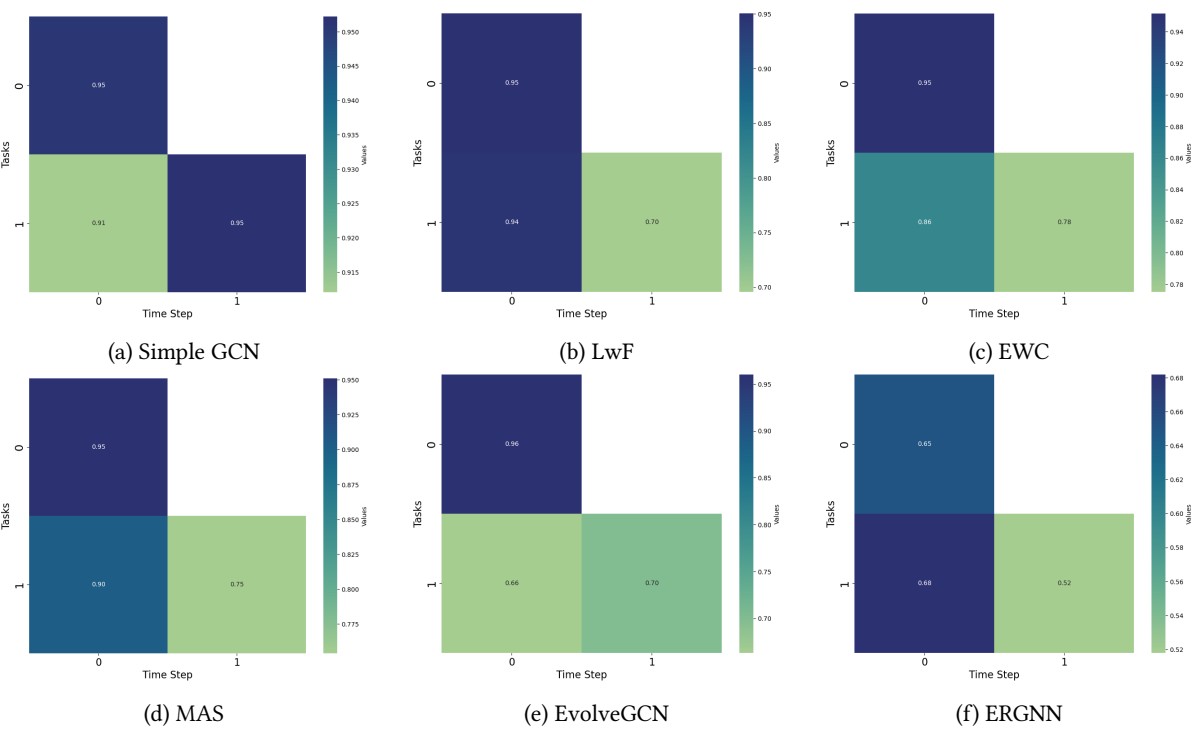

Figure 14: Visualization of the performance matrix of the methods in CLASS-IL SETTING on dataset DBLP

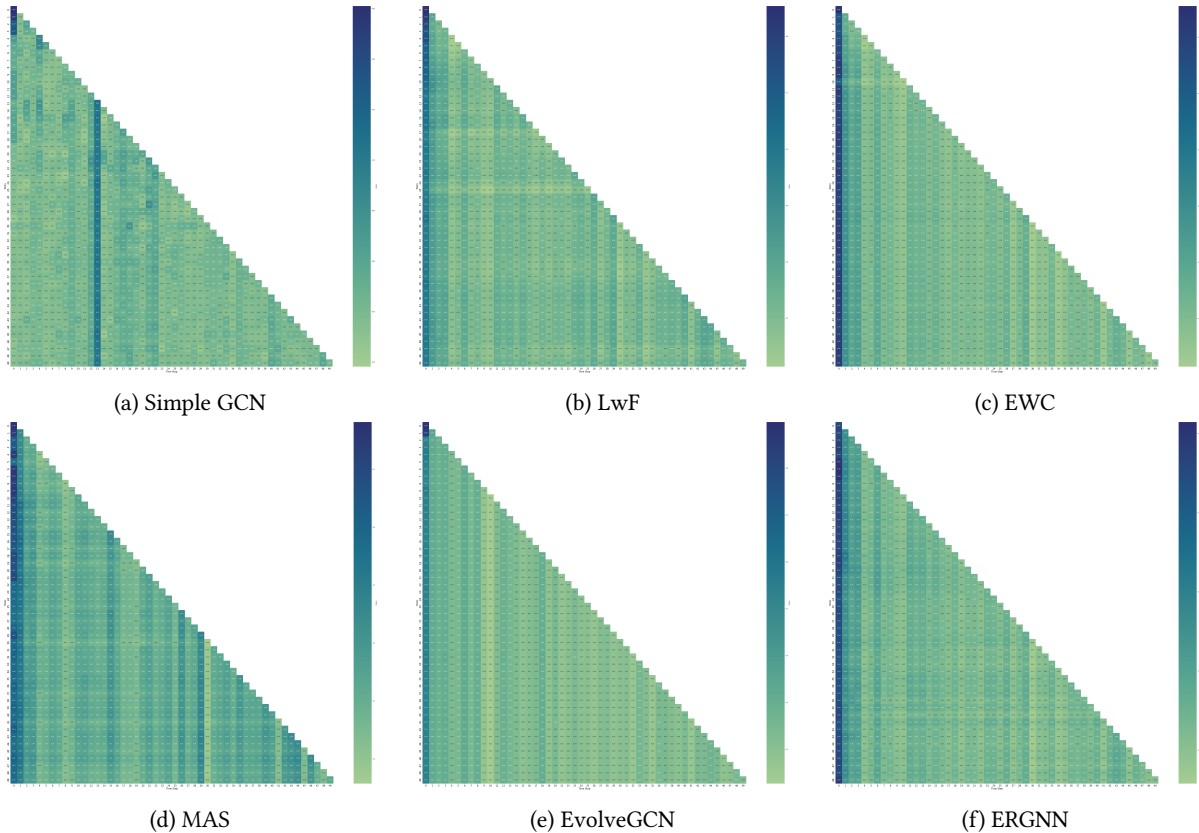

Figure 15: Visualization of the performance matrix of the methods in CLASS-IL SETTING on dataset Yelp

