# OpenReview forum: "AGALE: A Graph-Aware Continual Learning Evaluation Framework"
_TMLR — Accepted by TMLR_

### Review · Reviewer_p3Zp · 2024-01-22

**Summary Of Contributions:**

The paper introduces the AGALE framework, which aims to address the challenges of evaluating continual graph learning in both single-labeled and multi-labeled scenarios. It fills gaps in the current literature by defining two generalized incremental settings for multi-label node classification tasks and developing new data split algorithms for curating CGL datasets. The authors conducted extensive experiments to evaluate and compare the performance of methods from continual learning, dynamic graph learning, and continual graph learning.

Furthermore, the paper presents performance comparisons of baseline models in the Task-IL setting, reporting Average Precision and average forgetting metrics. It also includes visual representations, such as the node degree distribution in subgraphs generated from the PCG dataset.

In conclusion, the AGALE framework contributes to the continual graph learning literature by providing a comprehensive evaluation framework that considers the challenges posed by multi-labeled scenarios. The paper's theoretical and empirical analyses shed light on the differences between single-label and multi-label cases, paving the way for the development of more effective models in continual graph learning.

Overall, the paper offers valuable insights and important contributions to the field of continual graph learning, and its findings are likely to inspire further research in this domain.

**Audience:**

Yes

**Claims And Evidence:**

Yes

**Requested Changes:**

- Give a more thorough literature review.

- give time complexity analyis

**Strengths And Weaknesses:**

Pros:
- The paper addresses the challenges of evaluating continual graph learning in both single-labeled and multi-labeled scenarios, filling gaps in the current literature.
- The AGALE framework defines two generalized incremental settings for multi-label node classification tasks and develops new data split algorithms for curating CGL datasets.
- The paper presents extensive experiments comparing methods from continual learning, dynamic graph learning, and continual graph learning, providing valuable insights into the performance of these methods in different scenarios.


Cons:
- The paper does not provide a detailed comparison of the AGALE framework with other existing evaluation frameworks for continual graph learning.
- The literature review is not sufficient in the literature review. For examples, some representative dynamic GNN is not included in the discussion of related works, such as:
DyExplainer: Explainable Dynamic Graph Neural Networks. WSDM'24.
NetWalk: A Flexible Deep Embedding Approach for Anomaly Detection in Dynamic Networks. KDD'18.
Link Prediction with Spatial and Temporal Consistency in Dynamic Networks.  IJCAI'17.
Spatio-Temporal Attentive RNN for Node Classification in Temporal Attributed Graphs. IJCAI'19.

- The time complexity analyis is suggested to be included and the experiemnts on it are also good for the evaluation.

---

> ### Author Response · Authors · 2024-03-19
> **Answers to reviewer Reviewer p3Zp**
>
> We thank the reviewer for the comments. Below we address each of the points raised by the reviewer and are happy to provide any further details if required. We mark all the changes in blue in the paper.
>
>
> **Comparision with existing frameworks.** In Section 3.4, on page 8, we presented a comprehensive comparison between AGALE and previous evaluation frameworks.  Specifically, we summarize the differences  of AGALE to previous frameworks with respect to integrating the multi-label scenario, mitigating information loss, averting data leakage across disparate time steps and within the same subgraph, and ensuring fair class splits. Concrete examples are also provided in the Introduction section to argue about the invalidity of previous frameworks for the multi-label graph data.
>
>
> **Literature review.** We added the discussion of the mentioned literature in the section 4.3 on page 11, the added contents are marked in blue. These papers focus on addressing challenges related to dynamic networks by incorporating temporal dynamics and spatial dependencies into graph neural network models, aiming to enhance tasks such as anomaly detection, link prediction, and node classification in dynamic graph structures.
>
>
> **Time complexity analysis.** this is indeed an important point. We add theoretical time and space complexity analysis in the Appendix A.3 on page 22, along with experiments on runtime for a single split.

---

### Review · Reviewer_zeky · 2024-02-20

**Summary Of Contributions:**

This paper points out the limitation of existing graph continual learning evaluation methods in that they focus mainly on single-label scenarios where each node has at most one associated label. To tackle this, the authors study more challenging and realistic scenarios where nodes in the graph can have multi-labels (with two different settings of task-incremental and class-incremental) and then propose simple yet effective graph partitioning algorithms to validate existing methods over those two continual learning settings. The authors perform extensive experiments and analyses with existing (graph) continual learning approaches, showcasing their advantages and limitations in the proposed challenging continual learning scenarios with multi-label nodes.

**Audience:**

Yes

**Broader Impact Concerns:**

I don't see any concerns about the ethical implications of this work.

**Claims And Evidence:**

Yes

**Requested Changes:**

Please see the weaknesses above.

**Strengths And Weaknesses:**

### Strengths
* This work tackles very important and challenging scenarios of graph continual learning where nodes in the graph can have multi-labels.
* The proposed graph partitioning algorithms have advantages over previous ones, for validating the methods on graph continual learning settings.
* The authors perform extensive experiments with multiple analyses, faithfully showing the efficacy of the proposed graph continual learning evaluation framework.
* This paper is well-written and very easy to follow.


### Weaknesses
* The claim that the proposed graph partitioning algorithms can minimize the loss of the graph topological structure is not very convincing. While they can ensure fair splits across different classes, it does not ensure that the subgraphs after splits can maintain (or preserve) the original graph topology.
* In the realistic scenario, some labels of certain nodes (e.g., users over the social network do not further show interest in certain topics and unfollow them) may be removed in the future, and considering (or at least discussing) this scenario along with the proposed evaluation framework would be worthwhile.

---

> ### Author Response · Authors · 2024-03-19
> **Answers to reviewer Reviewer zeky**
>
> We thank the reviewer for the comments. Below we address each of the points raised by the reviewer and are happy to provide any further details if required. We mark all the changes in blue in the paper.
>
> **Graph partitioning algorithm.** The reviewer is totally correct on this point. The splits may not always be reflective of the original graph topology. Our strategy instead leads to the minimization of the loss of graph edges. For instance, in previous strategies, a certain node may never appear at any of the time points (see Figure 2 and the corresponding explanation). The removal of such specific nodes may lead to isolated nodes.  In particular, with AGALE, the multi-labeled nodes are either present with a subset of their labels (Task-Il setting) or growing set up to their complete label set(Class-IL setting).
>
>
> **Efficient forgetting.** It is a fair comment. What is described in this comment is effective forgetting of the past while still avoiding catastrophic forgetting, which is interesting to consider in the future. We added the corresponding discussion in section 7, Conclusion, on page 18.

---

> > ### Comment · Reviewer_zeky · 2024-03-21
> >
> > Thank you for your response.

---

### Review · Reviewer_KrEX · 2024-03-06

**Summary Of Contributions:**

The paper suggests new benchmarks for graph continual learning. The paper lists different continual graph settings and shows which of the current evaluation methods are more suited for which settings -- highlighting disentangles in all of them. Then, it offers a graph-aware evaluation framework. Afterward, the paper conducts an empirical study, comparing different algorithms on different datasets using the suggested evaluation method. The paper shows new insights gained on these datasets by using AGALE

**Audience:**

Yes

**Claims And Evidence:**

Yes

**Requested Changes:**

The major problem in my opinion is the writing. Once the paper is made more concise, I'm leaning toward acceptance, as its strengths outweigh its weaknesses.

**Strengths And Weaknesses:**

Strengths:
- The method is simple and easy to understand and follow. The intuition behind it is strong, and the problems it raises in the existing literature are important.
- The empirical section is very methodological, checking a multitude of datasets and algorithms, suggesting new insights using AGALE
- Although simplistic, the suggested method in the novel to the best of my knowledge

Weaknesses:
- The writing could be significantly more concise. Many points repeat themselves and are overly wordy. For example, the actual new ideas of the paper are introduced first on page 6, which is problematic. Section 2 contains many notations from previous art that are never used later on. Section 4 repeats many of the points made in Section 1 (specifically, 4.4 is very repetitive). Many subsections could have been shortened to just a few sentences.
- The significance of the paper is marginal -- it suggests an evaluation for a very specific case: Graph data + continual learning + multi-label. While I understand that this can not be addressed by the authors, it is still a weakness of the work.

---

> ### Author Response · Authors · 2024-03-19
> **Answers to reviewer KrEX**
>
> We thank the reviewer for the comments. Below, we address each of the points raised by the reviewer and are happy to provide any further details if required. We mark all the changes in blue in the paper.
>
>
> **Writing.** We specifically updated the section 1 introduction, rewrote the section 3.4 (originally section 4.4) to remove the repetitive parts, and related work is moved to section 4. We believe that a more elaborate discussion on the different strategies (as done in related work) to achieve Continual learning would help the reader to follow our result analysis as well as the time/space complexity analysis.
>
>
> **Significance of the paper.** We would like to point out while we illustrate the issues with current frameworks using a specific case of multi-label learning, our developed framework is more general and applicable for multi-label and multiclass node classification tasks ( please also see experiments on multi-class datasets in Appendix A.2.) as well as easily extendable to consider the case of graph classification and edge classification as illustrated in Section 3.4.. With graph-based learning being applied to many domains and fields we believe that our evaluation framework would encourage development of more general methods taking into account of the complexities of the multi-label scenario.

---

### Author Response · Authors · 2024-06-03
**Camera Ready Version Uploaded**

Dear action editor,

thank you again for your time through the reviwing process and we would like to inform you that the camera-ready version of our manuscript has been uploaded.

Best regards,

 authors

---

### Decision · Action_Editor_wtjA · 2024-05-05

**Recommendation:** Accept as is

**Comment:**

The paper wants to improve and standardize evaluation of countinual graph learning. The paper lists different continual graph settings and offers a graph-aware evaluation framework. The reviewers were leaning positive but had some concerns about writing and related works. In the revised version post rebuttals most of these concerns were resolved making the paper ready for publication.

**Audience:**

Yes, a good portion of graph learning community (both deep and classical) can be interested in this evaluation framework.

**Claims And Evidence:**

Mainly an evaluation paper. Provides underlying framework code and a number of baselines.